# Disordered proteins interact with the chemical environment to tune their protective function during drying

Shraddha KC[1], Kenny H Nguyen[1], Vincent Nicholson[1], Annie Walgren[1], Tony Trent[1], Edith Gollub[2], Paulette Sofia Romero-Perez[2], Alex S Holehouse[3,4], Shahar Sukenik[2]*, Thomas C Boothby[1]*

[1]Department of Molecular Biology, University of Wyoming, Laramie, United States; [2]Department of Chemistry and Biochemistry, University of California Merced, Merced, United States; [3]Department of Biochemistry and Molecular Biophysics, Washington University School of Medicine, St Louis, United States; [4]Center for Biomolecular Condensates, Washington University in St. Louis, St. Louis, United States

## eLife assessment

This **important** study investigates the sensitivity to endogenous cosolvents of three families of intrinsically disordered proteins involved with desiccation. The findings, drawn from well-designed experiments and calculations, suggest a functional synergy between sensitivity to small molecule solutes and convergent desiccation protection strategy. The evidence is found to be **convincing**, and the authors provide appropriate caveats since the study's conclusions are based on a small number of proteins. This work will be of interest to biochemists and biophysicists interested in the conformation–function relationship of intrinsically disordered proteins.

*For correspondence:
ssukenik@ucmerced.edu (SS);
tboothby@uwyo.edu (TCB)

**Abstract** The conformational ensemble and function of intrinsically disordered proteins (IDPs) are sensitive to their solution environment. The inherent malleability of disordered proteins, combined with the exposure of their residues, accounts for this sensitivity. One context in which IDPs play important roles that are concomitant with massive changes to the intracellular environment is during desiccation (extreme drying). The ability of organisms to survive desiccation has long been linked to the accumulation of high levels of cosolutes such as trehalose or sucrose as well as the enrichment of IDPs, such as late embryogenesis abundant (LEA) proteins or cytoplasmic abundant heat-soluble (CAHS) proteins. Despite knowing that IDPs play important roles and are co-enriched alongside endogenous, species-specific cosolutes during desiccation, little is known mechanistically about how IDP-cosolute interactions influence desiccation tolerance. Here, we test the notion that the protective function of desiccation-related IDPs is enhanced through conformational changes induced by endogenous cosolutes. We find that desiccation-related IDPs derived from four different organisms spanning two LEA protein families and the CAHS protein family synergize best with endogenous cosolutes during drying to promote desiccation protection. Yet the structural parameters of protective IDPs do not correlate with synergy for either CAHS or LEA proteins. We further demonstrate that for CAHS, but not LEA proteins, synergy is related to self-assembly and the formation of a gel. Our results suggest that functional synergy between IDPs and endogenous cosolutes is a convergent desiccation protection strategy seen among different IDP families and organisms, yet the mechanisms underlying this synergy differ between IDP families.

## Introduction

Intrinsically disordered proteins (IDPs) make up about 40% of the eukaryotic proteome (*Ward et al., 2004*; *Holehouse and Kragelund, 2024*). Unlike typical well-folded proteins, IDPs are characterized by a lack of defined tertiary structure, and instead exist as an ensemble of dynamic, interconverting conformations (*van der Lee et al., 2014*; *Mao et al., 2013*). Despite their disordered nature, IDPs are known to play important roles in many biological processes, including regulation of transcription and translation, metabolic signaling, subcellular organization, molecular chaperoning, and response and adaptation to environmental cues (*van der Lee et al., 2014*; *Kedia et al., 2017*).

Despite lacking a stable three-dimensional structure, IDPs still follow a similar paradigm by which form begets function. Different from well-folded proteins, however, an IDP's sequence determines the ensemble of conformations it adopts, and this ensemble can be important for the IDP's function(s) (*Holehouse and Kragelund, 2024*; *Mao et al., 2013*; *Das et al., 2015*). However, sequence is not the only determinant of the conformations present in an IDP's ensemble (*Soranno et al., 2014*; *Moses et al., 2020*; *Moses et al., 2023*). This is because IDP ensembles have relatively few intramolecular bonds and a large solvent-accessible surface area, which makes their ensembles more sensitive to the physicochemical environment than the relatively rigid structures of well-folded proteins (*Moses et al., 2020*; *Moses et al., 2023*; *Theillet et al., 2014*).

The sensitivity of IDP ensembles to their solution environment and their link to IDP function poses a fundamental question: how do sequence and solution combine to tune IDP ensemble and function? To explore this question, we focus on a biological phenomenon where the intracellular environment undergoes drastic physical chemical changes: desiccation.

Organisms across every biological kingdom can survive near-complete desiccation by entering a state of reversible suspended metabolism known as anhydrobiosis (from Greek for 'life without water') (*Crowe, 2014*; *Hesgrove and Boothby, 2020*). As water effluxes from the cell during drying, the concentration of cosolutes increases by orders of magnitude, dramatically changing the physicochemistry of the cell (*Romero-Perez et al., 2023*; *Oliver et al., 2020*). In addition to the decrease in water content and concomitant increase in cosolute concentrations, the composition of the intracellular environment changes massively because of a regulated metabolomic response to drying mounted by anhydrobiotic organisms (*Hesgrove and Boothby, 2020*; *Koster and Leopold, 1988*).

The acquisition of desiccation tolerance has historically been linked to the intracellular buildup of cosolutes such as trehalose, sucrose, arabinose, stachyose, and raffinose in plants and trehalose in some animals, fungi, and bacteria (*Koster and Leopold, 1988*; *Angeles-Núñez and Tiessen, 2010*; *Fait et al., 2006*; *Ryabova et al., 2020*; *Nguyen et al., 2022*; *Tapia and Koshland, 2014*). More recently, the accumulation of high levels of IDPs has also been linked to desiccation tolerance in many organisms (*Hesgrove and Boothby, 2020*; *Hincha and Thalhammer, 2012*; *Boothby et al., 2017*; *Hernández-Sánchez et al., 2022*; *Hand et al., 2011*). Common examples of desiccation-related IDPs include the late embryogenesis abundant (LEA) proteins, which are the most widely studied desiccation-related IDPs due to their early identification and widespread distribution among different species and kingdoms of life (*Hincha and Thalhammer, 2012*; *Hernández-Sánchez et al., 2022*; *Hand et al., 2011*; *Hundertmark and Hincha, 2008*). LEA proteins are classified into seven different families based on the presence of conserved motif sequences (*Hundertmark and Hincha, 2008*; *Artur et al., 2019*). Another family of desiccation-related IDPs are the tardigrade-specific cytoplasmic abundant heat-soluble (CAHS) proteins (*Hesgrove and Boothby, 2020*; *Boothby et al., 2017*; *Yamaguchi et al., 2012*).

Simultaneous enrichment of disordered proteins and endogenous cosolutes during desiccation promotes an ideal setting in which to study IDP-cosolute interactions (*Nguyen et al., 2022*; *Goyal et al., 2005*; *Kim et al., 2018*; *LeBlanc and Hand, 2021*). In the desiccation field, these interactions have been observed to produce a functional synergy in promoting tolerance to drying. Trehalose, a cosolute enriched alongside IDPs in many desiccation-tolerant systems, has previously been shown to enhance IDP protective function in vitro and in vivo (*Nguyen et al., 2022*; *Goyal et al., 2005*; *Kim et al., 2018*; *LeBlanc and Hand, 2021*). These observations prompted us to ask about the specificity of these interactions and if desiccation-related IDPs may have coevolved to work synergistically alongside their endogenously enriched cosolutes to promote desiccation protection.

To test whether desiccation-related IDP sequences have evolved to work with their intracellular chemical environment, here we use representative proteins from three families of IDPs: one CAHS

protein (CAHS D) and proteins from two LEA families. Our result shows that full-length CAHS D and LEA proteins derived from four different organisms synergize better with endogenous protective cosolutes compared to protective exogenous cosolutes from other organisms.

To reveal the underpinnings of cosolute:IDP synergy, we examine the secondary and tertiary structure of protective IDPs in the presence of two disaccharides that are similar in terms of size but distinct with respect to chemistry and use across taxa. In all cases, the secondary structure (residual helicity) and tertiary structure (radius of gyration) do not change significantly in the presence of synergistic cosolutes and thus cannot explain the enhancement in function observed with synergistic cosolutes. We next assessed quaternary structure as both CAHS and LEA proteins are known to oligomerize (*Hernández-Sánchez et al., 2022*; *Sanchez-Martinez et al., 2023*), and for CAHS proteins, oligomerization leads to gelation (*Sanchez-Martinez et al., 2023*; *Eicher et al., 2022*; *Malki et al., 2022*; *Yagi-Utsumi et al., 2021*; *Tanaka et al., 2022*). While synergistic cosolutes did not influence LEA oligomerization, CAHS D oligomerization and gelation were enhanced in the presence of synergistic cosolutes. We further show that CAHS D's synergy can be explained through direct repulsive interactions between cosolutes and CAHS D's side chains. However, this explanation does not hold for LEA proteins, implying that synergy in different protein families occurs through distinct mechanisms.

Our study showcases that different families of protective IDPs can have orthogonal modes of action and different functions in divergent solution environments. Beyond expanding our understanding of desiccation tolerance, these findings shed light on the sensitivity of IDP ensemble and function to the chemical composition of their environment. This is important as IDPs are ubiquitous across biology and function in key developmental processes and disease states that are concomitant with large changes in intracellular chemistry. Understanding how disordered proteins interact and evolve with the solution environment will provide insights into these biological mechanisms and phenomena.

## Results

### Desiccation-related IDPs are enriched in organisms alongside specific cosolutes during drying

To test whether IDP sequences have evolved to be functionally tuned by the composition of the intracellular environment during drying, we selected six desiccation-related IDPs. These IDPs come from two LEA families (*Hundertmark and Hincha, 2008*; *Battaglia et al., 2008*) as well as the CAHS family (*Table 1*). We selected four LEA_4 proteins each from a different desiccation tolerant organism. These organisms include the plant *Arabidopsis thaliana* (AtLEA3-3), the nematode *Aphelenchus avenae* (AavLEA1), the tardigrade *Hypsibius exemplaris* (HeLEA68614), and the rotifer *Adineta vaga* (AvLEA1C) (*Table 1*). To assess whether synergy extends across LEA families, we selected a LEA_1 protein from *A. thaliana* (AtLEA4-2) (*Table 1*). Finally, CAHS D was selected from the tardigrade *H. exemplaris* (*Table 1*). These organisms were selected not only because they all utilize LEA proteins to survive desiccation, but also because they accumulate different disaccharides to varying degrees during drying (*Angeles-Núñez and Tiessen, 2010*; *Fait et al., 2006*; *Nguyen et al., 2022*; *Higa and Womersley, 1993*; *Browne et al., 2004*; *Hengherr et al., 2008*; *Tunnacliffe and Lapinski, 2003*; *Lapinski and Tunnacliffe, 2003*; *Table 1*). The organisms we chose use one or both of two disaccharides – trehalose and sucrose which are similar in size, but chemically distinct (*Table 1*).

### LEA motifs are not sufficient to mediate synergistic interaction with endogenous cosolutes during desiccation

To assess whether endogenous cosolutes induce functional changes in desiccation-related IDPs, we began by testing peptides encoding LEA motifs derived from full-length LEA_4 and LEA_1 proteins (*Table 1*). Family 1 LEA (LEA_1) proteins are characterized by a 20-mer repeating motif, whereas Family 4 LEA (LEA_4) proteins are characterized by the repetition of an 11-mer LEA_4 motif (*Hundertmark and Hincha, 2008*; *Battaglia et al., 2008*; *Dure L 3rd, 1993*; *Knox-Brown et al., 2020*; *Furuki and Sakurai, 2016*). These motifs are often found in multiple linear or nonlinear repeats across the length of a LEA protein (*Hundertmark and Hincha, 2008*; *Battaglia et al., 2008*; *Dure L 3rd, 1993*). Interestingly, LEA_4 motifs have previously been suggested to be sufficient to confer desiccation protection to desiccation-sensitive proteins and membranes to a degree similar to full-length LEA proteins, both in vitro and in vivo (*Furuki and Sakurai, 2016*; *Furuki et al., 2012*; *Furuki and Sakurai,*

**Table 1.** Summary of organisms, disaccharides, and proteins used in this study.

Table displaying the organismal source of representative LEA_4, LEA_1, and CAHS proteins used in this study. In addition, the table displays endogenous cosolute reported in the literature to be co-enriched alongside late embryogenesis abundant (LEA) and cytoplasmic abundant heat-soluble (CAHS) proteins during desiccation in the given organism. The consensus sequence of 11-mer LEA_4 or 20-mer LEA_1 motifs as well as the length of the full-length proteins and predicted disorder using Metapredict (*Emenecker et al., 2021*) are shown. Shaded areas in the disorder plot correspond to the motif coordinates in the full-length LEA proteins. We note that the reason many of these profiles contain large folded regions is because the amphipathic LEA and CAHS proteins are predicted to form helices, which metapredict infers and incorrectly highlights these regions as 'folded' when really they are disordered in isolation.

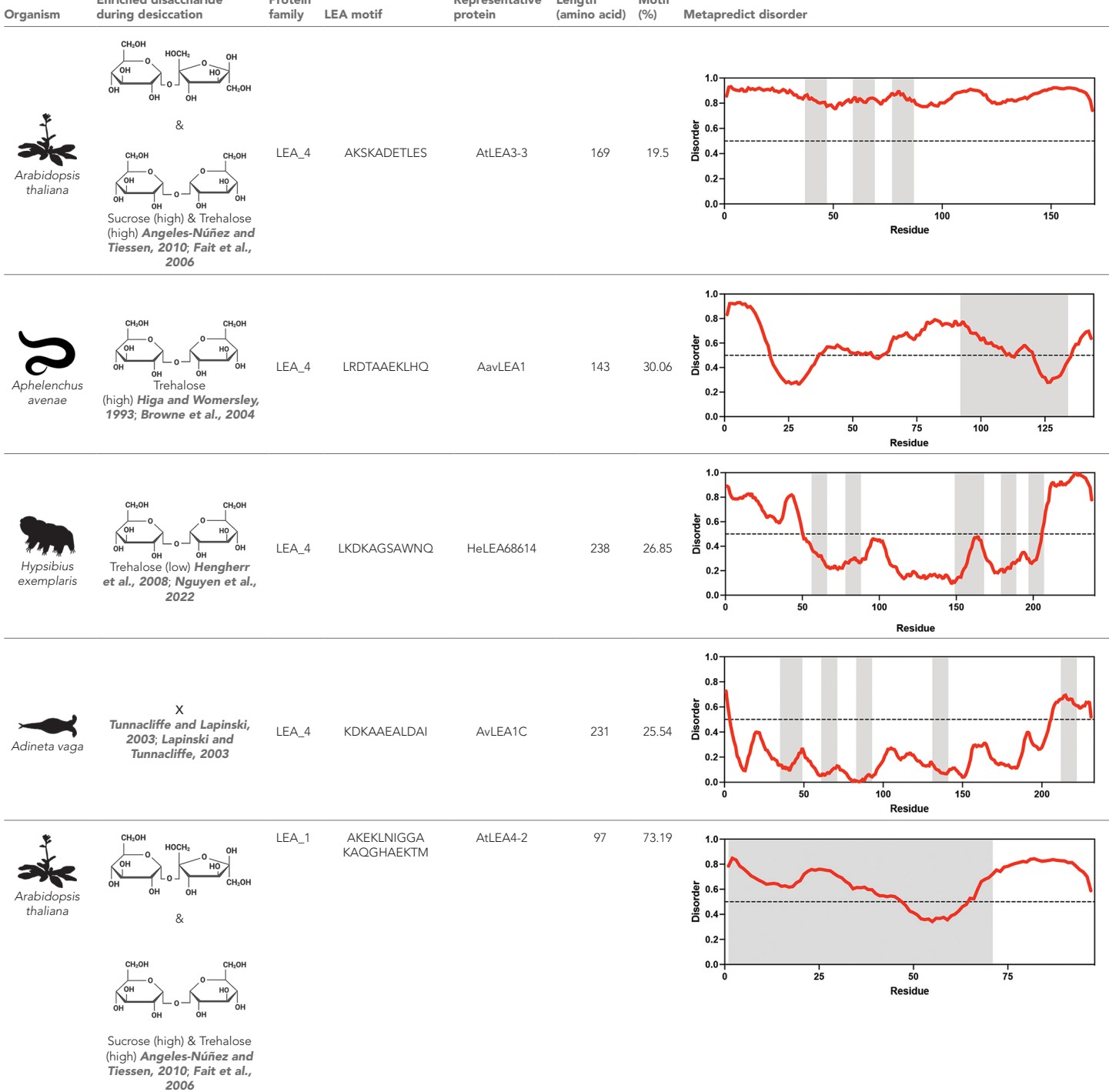

| Organism | Enriched disaccharide during desiccation | Protein family | LEA motif | Representative protein | Length (amino acid) | Motif (%) | Metapredict disorder |
|---|---|---|---|---|---|---|---|
| *Arabidopsis thaliana* | Sucrose (high) & Trehalose (high) *Angeles-Núñez and Tiessen, 2010*; *Fait et al., 2006* | LEA_4 | AKSKADETLES | AtLEA3-3 | 169 | 19.5 | |
| *Aphelenchus avenae* | Trehalose (high) *Higa and Womersley, 1993*; *Browne et al., 2004* | LEA_4 | LRDTAAEKLHQ | AavLEA1 | 143 | 30.06 | |
| *Hypsibius exemplaris* | Trehalose (low) *Hengherr et al., 2008*; *Nguyen et al., 2022* | LEA_4 | LKDKAGSAWNQ | HeLEA68614 | 238 | 26.85 | |
| *Adineta vaga* | X *Tunnacliffe and Lapinski, 2003*; *Lapinski and Tunnacliffe, 2003* | LEA_4 | KDKAAEALDAI | AvLEA1C | 231 | 25.54 | |
| *Arabidopsis thaliana* | Sucrose (high) & Trehalose (high) *Angeles-Núñez and Tiessen, 2010*; *Fait et al., 2006* | LEA_1 | AKEKLNIGGA KAQGHAEKTM | AtLEA4-2 | 97 | 73.19 | |

*Table 1 continued on next page*

*Table 1 continued*

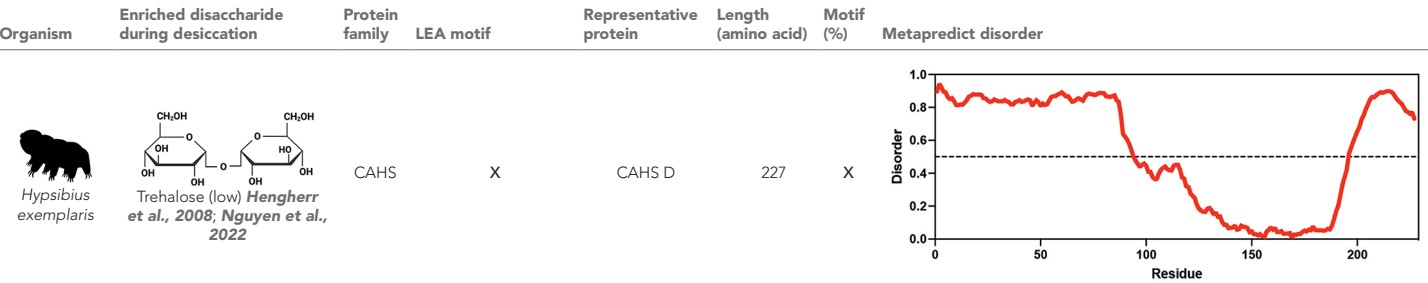

| Organism | Enriched disaccharide during desiccation | Protein family | LEA motif | Representative protein | Length (amino acid) | Motif (%) | Metapredict disorder |
|---|---|---|---|---|---|---|---|
| *Hypsibius exemplaris* | Trehalose (low) *Hengherr et al., 2008*; *Nguyen et al., 2022* | CAHS | X | CAHS D | 227 | X | |

The online version of this article includes the following source data for table 1:

**Source data 1.** LEA motif coordinates presented in *Table 1*.

*2014*; *Hibshman and Goldstein, 2021*). With this in mind, we expected to observe synergy between LEA motif repeats and their paired endogenous cosolute(s).

We generated 11-mer LEA_4 motif peptides (At11, Aav11, He11, and Av11) and a 20-mer LEA_1 peptide (At20) and measured the ability of these motifs to protect lactate dehydrogenase (LDH), a desiccation-sensitive enzyme during drying. LDH assay is used to assess the function of desiccation protectants to protect the activity of LDH, which otherwise retains only approximately 2% of its pre-desiccation activity when dried and rehydrated (*Nguyen et al., 2022*; *Boothby et al., 2017*; *Goyal et al., 2005*; *Piszkiewicz et al., 2019*).

The protective capacity for each LEA motif and cosolute was assessed across a range of concentrations with LDH (*Figure 1A*). Most LEA_4 motifs displayed levels of protection so low that a 50% level of protection could not be reached even at concentrations exceeding 1 mM (*Figure 1A and B*, *Figure 1—figure supplement 1A*). Additionally, higher concentrations of the LEA_4 motifs tend to inactivate the enzyme when kept under control conditions (4°C, see 'Materials and methods'), over a 16 hr incubation period during the assay (*Figure 1—figure supplement 1C*). The LEA_1 20-mer motif At20, however, showed robust concentration-dependent protection of LDH, demonstrating that LEA_4 and LEA_1 motifs are functionally distinct (*Figure 1A and B*, *Figure 1—figure supplement 1A*). Concentration-dependent protection was also observed for our cosolutes trehalose and sucrose (*Figure 1A*, *Figure 1—figure supplement 1A*).

Since LEA_4 motifs often exist in tandem repeats of 11-mers within a full-length LEA protein (*Furuki and Sakurai, 2016*; *Furuki et al., 2012*; *Shimizu et al., 2010*), we wondered if the observed lack of protection was a result of their short length or repeat number. We synthesized 2X (At22) and 4X (At44) tandem repeats of the *A. thaliana* 11-mer LEA_4 motif (At11). At22 and At44 show minimal potency in preserving in vitro LDH function during drying (*Figure 1A*, *Figure 1—figure supplement 1A*).

Despite the low protection displayed by our LEA_4 peptides, we opted to use them in cosolute synergy assays, as we reasoned that perhaps they would become functional when in solution with trehalose or sucrose. We picked suboptimal concentrations of protectants (*Supplementary file 1*) so that under instances of synergistic protection the additive protection of cosolute:peptide mixtures would not exceed 100%. We then performed synergy assays where sucrose or trehalose was combined with LEA motifs at molar ratios of 1:100, 1:10, 1:1, 10:1, and 100:1 (cosolute:protein). The upper limit (10:1 and 100:1) of these ratios closely aligns with known cosolute:protein ratios that produce synergistic protection between the tardigrade disordered protein CAHS D and trehalose (*Nguyen et al., 2022*). Here we report on synergy by showing the individual protective ability of the cosolute and IDP on its own, the sum of these protective values (hypothetical additive effect), and the actual measured protection produced by combination of the cosolute and peptide (*Figure 1C*). We quantify synergy using the following equation:

$$\% \text{ synergy} = (\% \text{ LDH protection})_{\text{IDP+cosolute}} - (\% \text{ LDH protection})_{\text{IDP}} - (\% \text{ LDH protection})_{\text{cosolute}}$$

In nearly all cases, synergy was not observed for 11-mer LEA_4 peptides with either sucrose or trehalose. In fact, in several cases, mixing LEA_4 peptides with sucrose or trehalose elicited antagonistic, rather than synergistic, effects (*Figure 1D and E*, *Figure 1—figure supplement 1B*). These results suggest that LEA_4 motifs do not robustly preserve LDH function, nor do they interact with

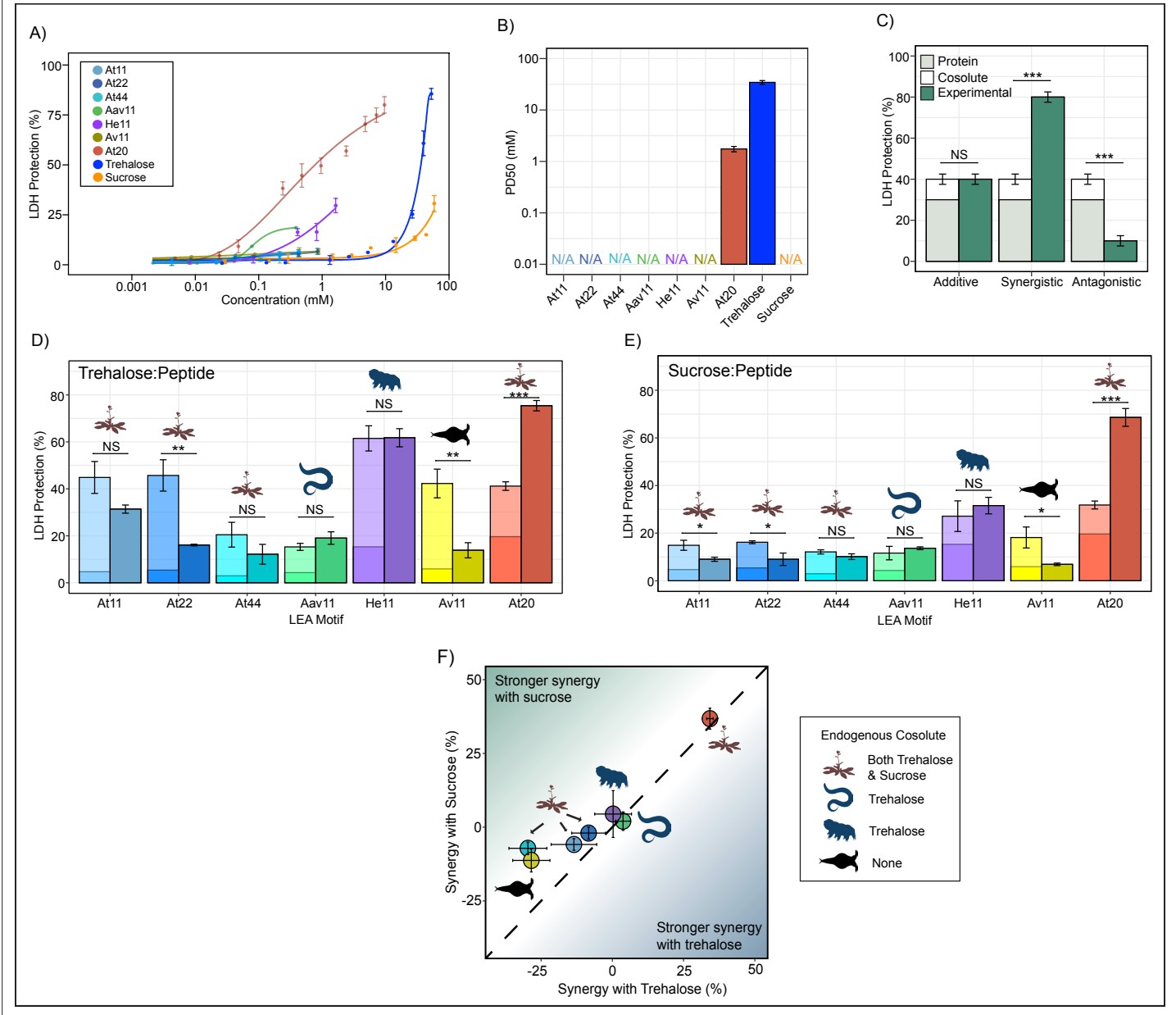

**Figure 1.** Late embryogenesis abundant (LEA) motifs are not synergistic with endogenous cosolutes. (**A**) Sigmoidal plot representing percent of lactate dehydrogenase (LDH) stabilization by LEA motifs and cosolutes as a function of the molar concentration. error bars = standard deviation. (**B**) Protective dose 50 (PD50) for additives obtained by sigmoidal fitting of data in (**A**). N/A represents instances where 50% protection was not achieved. (**C**) Example plot showing possible outcomes (additive, synergistic, or antagonistic effect) from cosolute:IDP mixtures. 'Cosolute' and 'Protein' represent the percent LDH protection by cosolute and protein respectively. Experimental represents the experimental protection resulting from cosolute and protein mixtures. (**D**) Synergy plots for trehalose:LEA peptide at 100:1 molar ratio. Welch's *t*-test was used for statistical comparison, error bars = standard deviation. (**E**) Synergy plots for sucrose:LEA peptide at 100:1 molar ratio. (**F**) Plot representing synergy with trehalose vs. sucrose from (**D, E**). Dotted line represents instances where there is equal synergy with both trehalose and sucrose. All LDH experiments were done with three technical replicates.

The online version of this article includes the following source data and figure supplement(s) for figure 1:

**Source data 1.** Lactate dehydrogenase (LDH) assay data for motifs and cosolutes used in the study related to *Figure 1A and B*.

**Source data 2.** Synergy data for cosolute:motif mixtures related to *Figure 1C–F*.

**Figure supplement 1.** Lactate dehydrogenase (LDH) assay for late embryogenesis abundant (LEA) motifs and cosolutes.

**Figure supplement 1—source data 1.** All data related to *Figure 1—figure supplement 1*.

cosolutes trehalose or sucrose in a functionally productive fashion. Similar to 11-mer motifs, the *A. thaliana* 22- and 44-mer peptides did not synergize with either trehalose or sucrose (*Figure 1D and E*, *Figure 1—figure supplement 1B*). Unlike LEA_4 motifs, the 20-mer LEA_1 motif, At20, was both protective and synergized with both trehalose and sucrose (*Figure 1D and E*, *Figure 1—figure supplement 1B*).

Taken together, these experiments demonstrate a diversity in disordered protein/motif function, where LEA_4 motifs largely are not protective to the enzyme LDH during drying, nor are they synergistic with endogenous cosolutes (*Figure 1F*). Conversely, the LEA_1 motif tested is highly protective and synergizes with either sucrose or trehalose (*Figure 1F*).

## Desiccation-related IDPs synergize with endogenous cosolutes

While LEA proteins are identified through homology in conserved LEA motif repeats, they also contain varying quantities of non-motif sequence (*Table 1*). Since we observed that LEA_4 motifs generally provide relatively little protection and tend not to synergize with endogenous cosolutes in LDH assays, we wondered if full-length LEA proteins might. We also included CAHS D in our analysis as it has been previously known to synergize with trehalose and, to a lesser extent, with sucrose (*Nguyen et al., 2022*).

We began by testing the baseline protection of our proteins using the LDH assay. All full-length LEA_1 and LEA_4 proteins confer protection for LDH activity up to the pre-desiccated value (*Figure 2A*, *Figure 2—figure supplement 1A*). Likewise, CAHS D provided concentration-dependent protection to LDH as previously observed (*Nguyen et al., 2022*; *Boothby et al., 2017*; *Piszkiewicz et al., 2019*; *Figure 2A*, *Figure 2—figure supplement 1A*). We also included bovine serum albumin (BSA) in these studies as a well-studied control (*Nguyen et al., 2022*; *Piszkiewicz et al., 2019*). Unlike the LEA motifs, most full-length proteins protected 50% LDH at concentrations less than 1 mM (*Figure 2B*).

Using data derived from the concentration range of LDH assays, we chose a suboptimal concentration that provides 15–45% protection for each protein to perform synergy experiments with (*Supplementary file 1*). Our results show that nearly all full-length IDPs showed synergy with either sucrose or trehalose or both (*Figure 2C and D*, *Figure 2—figure supplement 1B*). Exceptions to this are AvLEA1C, which is derived from a rotifer that accumulates neither trehalose nor sucrose, and BSA, which comes from cows, which of course have no capacity for anhydrobiosis. Remarkably, in cases where LEA proteins displayed synergy, they were always more synergistic with endogenous compared to exogenous cosolutes (*Figure 2E*). We also calculated synergy using two different approaches: one using equilibrium constant ($K_{app}$) as a metric and other using the Bliss independent model. Our results for synergy are largely invariant regardless of the method we use (*Figure 2—figure supplement 2A–F*).

Taken together, these experiments demonstrate that synergistic interactions between IDPs and cosolutes extend across multiple families of desiccation-related IDPs found in a variety of organisms. Our data shows that these six IDPs synergize best with their endogenous cosolute to promote desiccation tolerance and we speculate that this may apply to other desiccation-related IDPs. It is also of note that while many of the LEA motifs tested at the beginning of this study did not display synergy with trehalose or sucrose, corresponding full-length proteins did. Likewise, At20 showed synergy with both trehalose and sucrose, while full-length AtLEA4-2 was only synergistic with sucrose. Thus, not only do these experiments demonstrate that full-length LEA proteins synergize with their endogenous cosolute(s) (*Figure 2E*), but they suggest that this synergy, at least in part, is driven by sequence features beyond conserved motifs.

## Trehalose and sucrose do not elicit local ensemble changes to desiccation-related IDPs in solution or in the dry state under the conditions tested

We next wondered what mechanism(s) drive the functional synergy observed between desiccation-protective IDPs and endogenous cosolutes. We reasoned that functional synergy might be driven by cosolute-induced changes to the IDP ensemble. To test this, we first examined the secondary structure contained in the ensemble of LEA proteins and CAHS D using circular dichroism (CD) spectroscopy.

Each full-length protein was first assessed using CD in an aqueous state by itself (*Figure 3A*). All full-length LEA proteins displayed a single minimum at ~200 nm (*Figure 3A*, black), indicating that

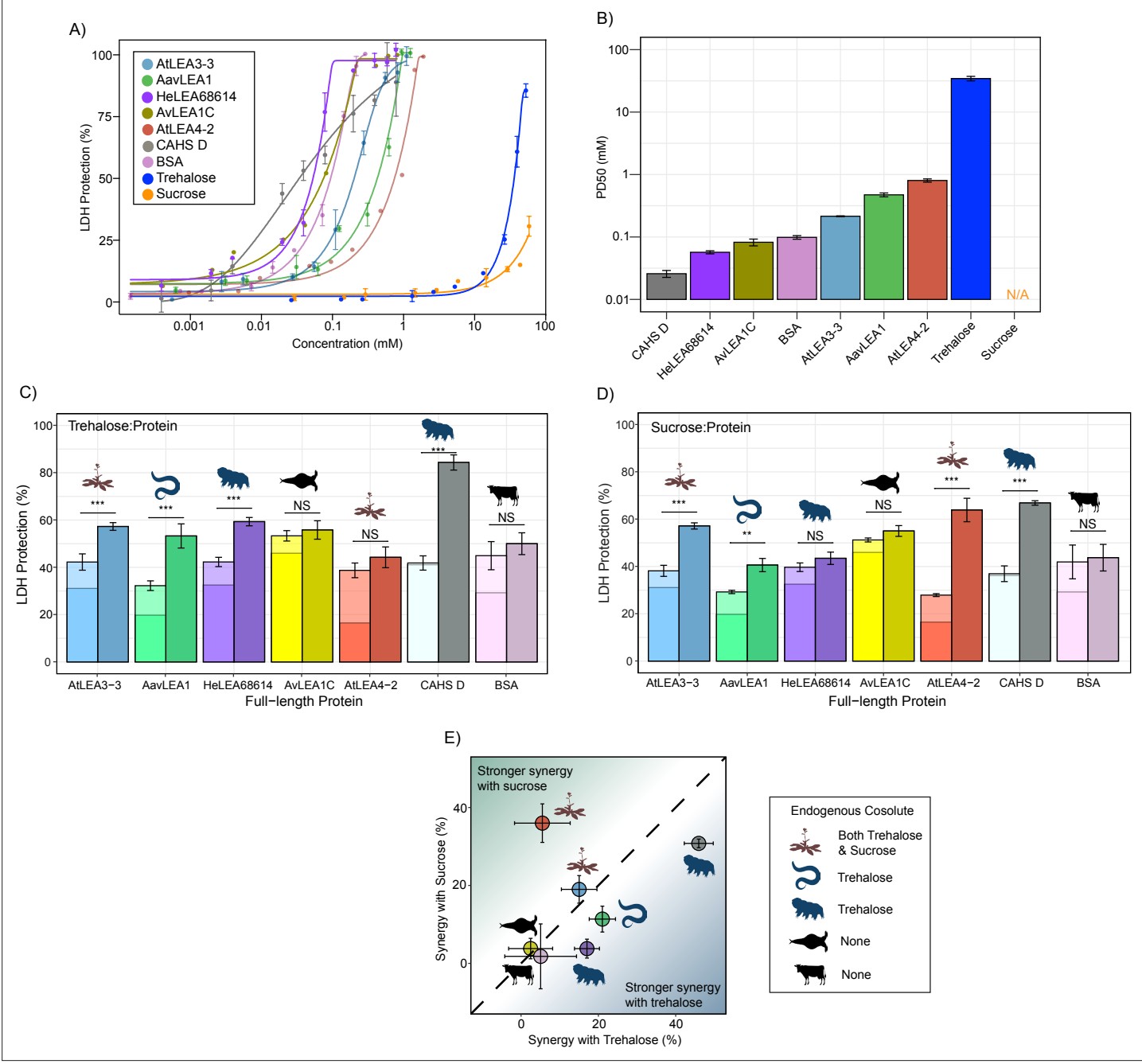

**Figure 2.** Full-length desiccation-related intrinsically disordered proteins (IDPs) act synergistically with cosolutes. (**A**) Concentration dependence of lactate dehydrogenase (LDH) protection by full-length proteins and cosolutes used in this study. (**B**) PD50 for additives obtained by sigmoidal fitting of concentration-dependent LDH protection from (**A**). N/A represents instances where 50% protection was not achieved. (**C**) Synergy plots for trehalose:protein at 100:1 molar ratio. Welch's t-test was used for statistical comparison, error bars = standard deviation. (**D**) Synergy plots for sucrose:protein at 100:1 molar ratio. Welch's t-test was used for statistical comparison, error bars = standard deviation. (**E**) Plot representing synergy with trehalose vs. sucrose from (**C, D**). All LDH experiments were done with three technical replicates.

The online version of this article includes the following source data and figure supplement(s) for figure 2:

**Source data 1.** Lactate dehydrogenase (LDH) assay data for full-length proteins and cosolutes used in the study related to *Figure 2A and B*.

**Source data 2.** Synergy data for cosolute:protein mixtures related to *Figure 1C–E*.

**Figure supplement 1.** Lactate dehydrogenase (LDH) assay for full-length proteins and cosolutes.

**Figure supplement 1—source data 1.** All data related to *Figure 2—figure supplement 1*.

*Figure 2 continued on next page*

*Figure 2 continued*

**Figure supplement 2.** Synergy calculation using alternative methods.

**Figure supplement 2—source data 1.** All data related to *Figure 2—figure supplement 2*.

LEA proteins are disordered in the aqueous state. CAHS D also displayed a minimum at ~200 nm and a slight minimum around ~220 nm, indicating that while disordered, it also has some propensity for helical structure in solution (*Figure 3A*, black), in line with previous studies (*Sanchez-Martinez et al., 2023*; *Wang et al., 2023*). This is in contrast to BSA, which showed a high propensity for helical structure as denoted by the double minima at 222 and 210 nm (*Figure 3A*, black). To see if the addition of cosolutes induces secondary structural change in the aqueous state, we obtained CD spectra of cosolute:protein mixtures at 100:1 molar ratios. Somewhat to our surprise, trehalose (*Figure 3A and C*, blue) and sucrose (*Figure 3A and C*, green) do not induce any significant structural changes to any of the full-length IDPs tested here (*Figure 3—figure supplement 1A*).

LEA proteins gain helical conformation upon drying or in response to low water availability. Drying-induced helicity has been postulated to drive their protective function (*Hernández-Sánchez et al., 2022*; *LeBlanc and Hand, 2021*; *Cuevas-Velazquez et al., 2016*). We reasoned that while our cosolutes do not induce detectable changes to LEA/CAHS secondary structure in solution, cosolutes could

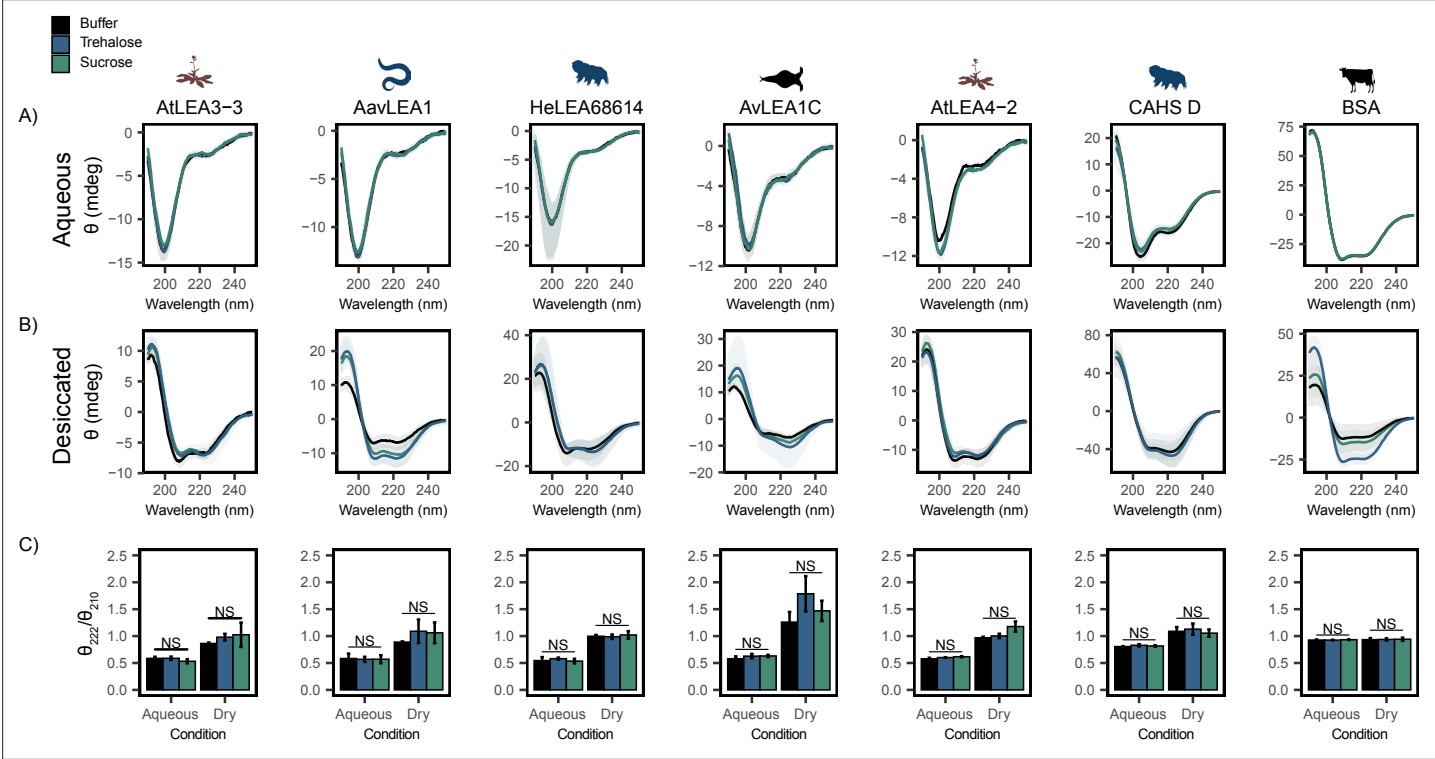

**Figure 3.** Functional synergy is not mediated by secondary structural changes. Buffer represents the circular dichroism (CD) analysis for the proteins without cosolutes. Trehalose and sucrose represent the CD analysis for trehalose:protein mixture and sucrose:protein mixture at 100:1 molar ratio respectively. (**A, B**) CD spectra for protein and cosolute:protein mixtures at 100:1 molar ratio under aqueous (**A**) and (**B**) desiccated conditions. Each plot represents the average of three technical replicates, with the shaded region representing the standard deviation of the average. (**C**) Changes in the ratio of CD signal at 222 and 210 nm for individual protein and cosolute:protein mixtures under aqueous and desiccated conditions. error bars = standard deviation, statistical analysis was done by pairwise *t*-test between each protein and its cosolute mixtures. Organismal icons that are colored blue indicate the species uses trehalose as an endogenous cosolute, brown indicates the species uses both trehalose and sucrose, and black indicates the species uses neither trehalose nor sucrose as an endogenous cosolute.

The online version of this article includes the following source data and figure supplement(s) for figure 3:

**Figure supplement 1.** Synergy and secondary structure.

**Figure supplement 1—source data 1.** All data related to *Figure 3—figure supplement 1*.

**Source data 1.** Circular dichroism (CD) data related to *Figure 3*.

induce structural changes in proteins in a dry state. To test this, we examined our proteins using CD in a desiccated state (*Bremer et al., 2017*; *Navarro-Retamal et al., 2016*). Our results show a significant structural change for all LEA proteins and CAHS D in the dry state, indicated by a shift from disordered spectra with a minimum at ~200 nm to a helical structure with two minima at ~222 and ~210 nm (*Figure 3B*). This is in contrast to BSA, which started out helical and showed little change in the spectrum (*Figure 3B*). These changes manifest for pure proteins without any addition of synergistic cosolutes. To quantify the influence of drying on secondary structure, we examined the changes in the ratio of CD signal at 222 and 210 nm. This ratiometric value reports on secondary structure in a concentration-independent way (*Greenfield, 2006*). Using this metric, all LEAs and CAHS D display a clear increase in helical propensity upon being desiccated (*Figure 3C*). On the other hand, the helical propensity of BSA remains very similar to its hydrated state, indicating that no dramatic structural change took place (*Figure 3C*).

Next, we examined combinations of our proteins with trehalose or sucrose in a desiccated state. As with aqueous samples, the addition of trehalose or sucrose did not induce significant changes in the secondary structure of LEA and CAHS D proteins in the dry state (*Figure 3B and C*, *Figure 3—figure supplement 1B*). To assess whether there is a link between the minimal structural changes we observed and functional synergy in LDH assays, the change in the ratio of signal at 210 and 222 nm in desiccated and aqueous states was compared to synergy observed for that same mixture. Synergistic protection observed in our LDH assays did not correlate with secondary structural changes with the addition of trehalose (p=0.973 for aqueous, p=0.2047 for desiccated, *Figure 3—figure supplement 1C and D*) or sucrose (p=0.6629 for aqueous, p=0.9763 for desiccated, *Figure 3—figure supplement 1E and F*).

Taken together, these results indicate that while the tested LEA proteins and CAHS D undergo a structural transition during desiccation, this phenomenon does not require, nor is it affected by, the presence of trehalose and sucrose. Furthermore, synergistic protection observed in our LDH assays is not mediated by local ensemble changes in these IDPs.

## Trehalose and sucrose do not elicit changes in global ensemble dimensions for desiccation-related IDPs under the tested conditions, but promote oligomerization of CAHS D

We next wondered if the synergistic interactions observed between our IDPs and cosolutes could instead be explained by a change in global dimensions, such as expansion or compaction of the protein. To measure global dimensions, which cannot be detected using CD, we used small-angle X-ray scattering (SAXS), which allows a model-free estimation of the radius of gyration ($R_g$) of an IDP as well as a prediction of the molecular weight (see 'Materials and methods'; *Kachala et al., 2015*). Each protein, at a concentration of 4 mg/mL, was measured with no cosolute and with different molar ratios of trehalose and sucrose. We reasoned that a cosolute-dependent change in the radius of gyration ($R_g$) could indicate changes in tertiary or quaternary structure, which may correlate with increased function or synergy. We note that for SAXS experiments cosolutes were used at concentrations between 20 and 50 mM due to technical restrictions on how much protein can accurately be assayed and a desire to maintain molar ratios used in other experiments.

We began by testing the $R_g$ of BSA in different solution conditions. As a well-folded protein, we expected that BSA would be relatively insensitive to changes in the solution environment. The $R_g$ values obtained via this approach match existing literature (*Figure 4A*; *Graewert et al., 2015*). Additionally, adding cosolutes did not modulate $R_g$ (*Figure 4A*). While our molecular weight approximations trended higher than expected for monomeric BSA, we reason that this may be due to the propensity of BSA to form small populations of low-level oligomers (*Babcock and Brancaleon, 2013*; *Levi and González Flecha, 2002*).

We next measured the $R_g$ of our LEA proteins in various solution conditions. We observed that regardless of the solution environment, all of our LEA proteins have an $R_g$ that falls within error of readings in other solution environments (*Figure 4B–F*). At the concentrations used here, cosolutes do not induce significant changes in the global dimensions of LEA proteins. While the predicted molecular weight (pMW) from SAXS for these proteins were somewhat variable, we see no consistent trend between the presence of cosolutes and change in pMW for any LEA protein (*Figure 4B–F*).

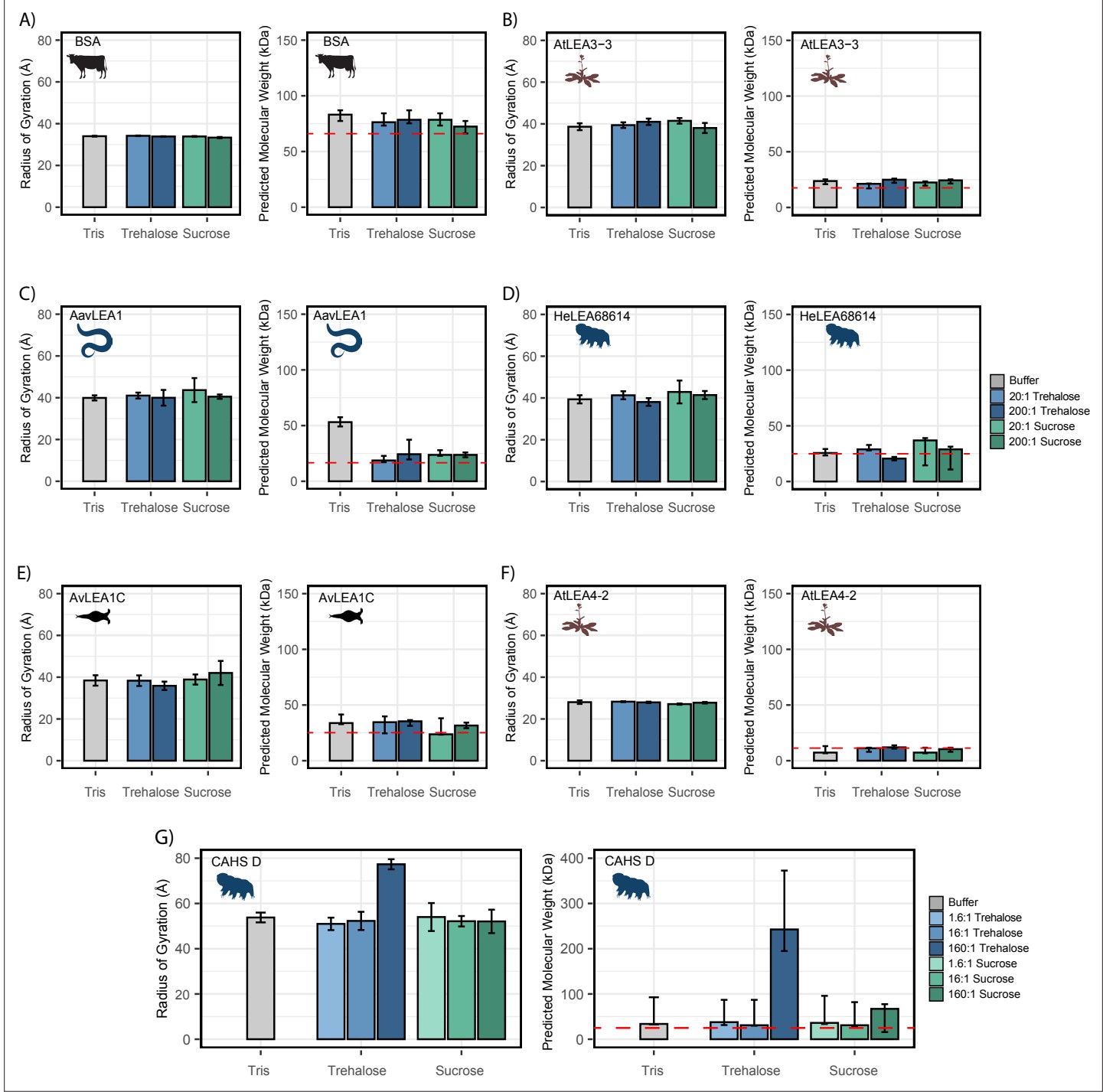

**Figure 4.** Cosolutes increase the global dimensions of CAHS D, but not late embryogenesis abundant (LEA) or bovine serum albumin (BSA). Analyzed data from small-angle X-ray scattering (SAXS) experiments of proteins at 4 mg/mL in 20 mM Tris-HCl pH 7. Proteins tested include (**A**) BSA (0.0578 mM), (**B**) AtLEA3-3 (0.22 mM), (**C**) AavLEA1 (0.249 mM), (**D**) HeLEA68614 (0.156 mM), (**E**) AvLEA1C (0.163 mM), (**F**) AtLEA4-2 (0.38 mM), and (**G**) CAHS D (0.156 mM). The left plot shows the radius of gyration of the protein in the presence of no cosolute (gray), increasing molar ratios of trehalose (blue shades), and increasing molar ratios of sucrose (green shades). All SAXS experiments were performed once with three technical replicates. Error bars represent uncertainty in the measurement, provided by BioXTAS RAW. The right plot shows molecular weight values derived from Guinier analysis (see 'Materials and methods'). The red dashed line indicates the monomeric protein's molecular weight. Color scheme is the same as in the left figure. Error bars represent >90% confidence interval, which is directly obtained from the analysis. Organismal icons that are colored blue indicate the species uses trehalose as an endogenous cosolute, brown indicates the species uses both trehalose and sucrose, and black indicates the species uses neither trehalose nor sucrose as an endogenous cosolute.

*Figure 4 continued on next page*

*Figure 4 continued*

The online version of this article includes the following source data and figure supplement(s) for figure 4:

**Source data 1.** Small-angle X-ray scattering (SAXS) data related to *Figure 4*.

**Figure supplement 1.** Raw small-angle X-ray scattering (SAXS) data from the experiments shown in *Figure 4* for 4mg/mL.

**Figure supplement 1—source data 1.** All data related to *Figure 4—figure supplement 1*.

**Figure supplement 2.** Photo-induced crosslinking of unmodified proteins (PICUP) gels for full-length late embryogenesis abundant (LEA) proteins and in mixtures with trehalose or sucrose.

**Figure supplement 2—source data 1.** Original raw files for all PICUP gels in *Figure 4—figure supplement 2*.

**Figure supplement 2—source data 2.** Files for PICUP gels in *Figure 4—figure supplement 2* and original gels with highlighted portions and labels.

Finally, we tested CAHS D in similar solution conditions. We obtained an $R_g$ value for CAHS D that lies between the values reported by other groups (see 'Materials and methods'; *Sanchez-Martinez et al., 2023*; *Malki et al., 2022*). While this $R_g$ was consistent in 1.6:1 disaccharide solutions, 16:1 disaccharide solutions, and 160:1 sucrose solution, we found that the 160:1 trehalose solution had a Guinier region that was sharply curved upward, even upon the protein's first exposure to the X-ray source (*Figure 4G*, *Figure 4—figure supplement 1G*). This is consistent with the presence of large oligomerized species. A Bayesian approximation of the molecular weight of these samples supported this result (*Figure 4G*). Most of our CAHS D samples showed a pMW that was only slightly elevated from the known value for the monomeric protein. In contrast, the 160:1 sucrose sample had a pMW about 300% higher than other samples, and the 160:1 trehalose sample had a pMW about 1000% higher (*Figure 4G*).

While SAXS shows little change in Rg or pMW for LEA proteins, there is evidence that not only CAHS D but also some LEA proteins tend to oligomerize (*Hernández-Sánchez et al., 2022*; *Shou et al., 2019*). However, LEA oligomerization appears to be weak and transient, requiring extreme crowding and/or sensitive methods to detect (*Hernández-Sánchez et al., 2022*; *Shou et al., 2019*; *Rivera-Najera et al., 2014*). We therefore wanted to use a more sensitive method to assess LEA oligomers and how they might be affected by cosolutes.

To characterize oligomerization of LEA proteins in a more sensitive fashion, we used photo-induced crosslinking of unmodified proteins (PICUP), a zero-length crosslinking method that uses a light-activatable crosslinking system and is known to capture transient oligomeric species (*Fancy and Kodadek, 1999*; *Preston and Wilson, 2013*). PICUP has previously been used to characterize oligomeric forms in LEA proteins in vitro (*Rivera-Najera et al., 2014*; *Liu et al., 2017*). All of our LEA proteins showed a propensity to form oligomers, even at low concentrations (*Figure 4—figure supplement 2*). However, the presence of trehalose or sucrose did not elicit changes in oligomeric populations (*Figure 4—figure supplement 2*). These results confirm that LEA proteins are able to form transient oligomers, but also demonstrate that at levels where sucrose and trehalose are synergistic with these proteins oligomerization is unaffected.

Taken together, these results show a divergence in the behavior of LEA and CAHS proteins in the presence of synergistic cosolutes. For CAHS D, the SAXS data suggests a relationship between the presence of cosolutes and increased oligomerization. However, this dataset showed no evidence of a cosolute inducible increase in molecular weight or $R_g$ for the five LEA proteins that we tested, at least at the concentrations used in this study. This was further supported by PICUP, which despite detecting LEA oligomers did not show that they were enhanced by cosolutes. Thus, our data suggests that CAHS D oligomerization is promoted by the presence of synergistic cosolutes while LEA oligomerization is not.

## Synergistic cosolutes promote gelation of CAHS D but not LEA proteins

CAHS D oligomers, as well as those of other CAHS proteins, have been reported to undergo self-assembly to form a gel network (*Sanchez-Martinez et al., 2023*; *Eicher et al., 2022*; *Malki et al., 2022*; *Yagi-Utsumi et al., 2021*; *Tanaka et al., 2022*; *Eicher et al., 2023*). We wondered if the cosolute-induced oligomerization of CAHS proteins in the presence of trehalose as seen in the SAXS experiments could be attributed to the propensity of CAHS D to form gels.

To test this, we performed differential scanning calorimetry (DSC) on CAHS D to observe the presence or absence of a gel melt. To begin, we tested CAHS D at 6 mg/mL (0.235 mM), which has previously been established to be a non-gelling concentration (*Sanchez-Martinez et al., 2023*). Consistent with this, at 0.235 mM, we find that CAHS D does not undergo a characteristic gel melt indicating a lack of gelation (*Figure 5A*, black). Addition of trehalose and sucrose at increasing molar ratios (1:1, 10:1, 100:1, and 500:1) showed thermal features characteristic of endothermic phase transitions (e.g., a gel melting), indicating that the presence of these cosolutes induced gelation (*Figure 5A*). Measuring the area under these melt curves allows us to calculate the enthalpy of melting (*Figure 5—figure supplement 1A*). Change in enthalpy measurements for the cosolute:protein mixtures relative to the protein provides us a quantification of how gelation is affected by different amounts of cosolutes. Trehalose induced significant gelation at a 100:1 ratio, while 500:1 of sucrose was required to induce a significant gel melt (*Figure 5B*). This is consistent with trehalose producing larger oligomeric species in our SAXS experiments (*Figure 4G*), indicating that trehalose has a larger influence than sucrose on the gelation of CAHS D (*Figure 5B*).

Furthermore, to test whether synergistic cosolutes enhance gelation, we tested CAHS D at 12 mg/mL (0.47 mM), a concentration above CAHS D's gelation threshold (*Sanchez-Martinez et al., 2023*). Addition of trehalose or sucrose at increasing molar concentrations promoted the formation of stronger gels evident by the enthalpy of melting measurements (*Figure 5—figure supplement 1B–E*). These experiments demonstrate that not only do trehalose and sucrose induce sub-gelling concentrations of CAHS D to form gels, but they also enhance the strength of gels formed by higher concentrations of the protein.

Unlike CAHS proteins, gelation of LEA proteins has not been commonly observed or reported. The exception to this is AfrLEA6 (a LEA protein from the brine shrimp *A. franciscana*) that appears to undergo phase separation, forming a hydrogel-like matrix upon desiccation (*Belott et al., 2020*). To see if our LEA proteins gel, we performed DSC experiments on our LEA proteins and cosolutes. We analyzed similar molar concentrations of LEA proteins on their own and in mixtures with cosolutes at equivalent ratios (1:1, 10:1, 100:1, and 500:1). None of our LEA proteins by themselves or in mixtures with trehalose or sucrose show evidence of gelation (*Figure 5B–G*). Likewise, BSA also failed to form a gel (*Figure 5H*).

Taken together, these results demonstrate that trehalose and sucrose affect the oligomerization and phase state of different IDPs in distinct ways. The observation that trehalose induces more synergy as well as more gelation of CAHS D relative to sucrose leads us to speculate that gelation of CAHS D and synergistic protection of LDH may be linked.

## Direct cosolute:IDP interactions drive synergy for CAHS D, but not LEA proteins

To explain the possible relationship between synergy and oligomerization-driven gelation of CAHS D, we quantified the interactions between each IDP and its cosolute environment using transfer free energies (TFEs). The TFE is a measure of the change in free energy undergone by a macromolecule when transferring from water to a concentrated solution of some osmolyte (typically 1 M) (*Tanford, 1970*; *Auton and Bolen, 2007*; *Auton and Bolen, 2005*). Using TFE values derived from literature, we calculated the effect of trehalose and sucrose on the ability of CAHS D to dimerize: $\Delta\Delta G_{tr}^{M\rightarrow D}$. This is calculated by finding the free energy of CAHS D's monomeric state upon transfer to an osmolyte solution ($\Delta G_{tr}^{M}$), doing the same for the dimeric state ($\Delta G_{tr}^{D}$), and then taking the difference (*Figure 6A*). A strong negative value for $\Delta\Delta G_{tr}^{M\rightarrow D}$ indicates that the presence of the cosolute pushes the population toward dimers. A positive value indicates that the addition of the cosolute pushes the population toward monomers.

In order to perform these calculations, we utilized AlphaFold2 and AlphaFold2 Multimer to determine plausible conformations for both CAHS D's monomeric and dimeric state. These structures have a helical linker region, which is consistent with our CD measurements (*Figure 3A and B*) and is consistent with previous reports for this protein (*Sanchez-Martinez et al., 2023*). We reasoned that dimerization is indicative of gelation since CAHS D dimers were especially prevalent in crosslinking data (*Sanchez-Martinez et al., 2023*), and previous research suggests that CAHS D dimer formation is a necessary step toward gelation (*Sanchez-Martinez et al., 2023*). TFEs were then calculated based on the solvent-accessible surface area of different chemical groups in monomeric vs. dimeric state

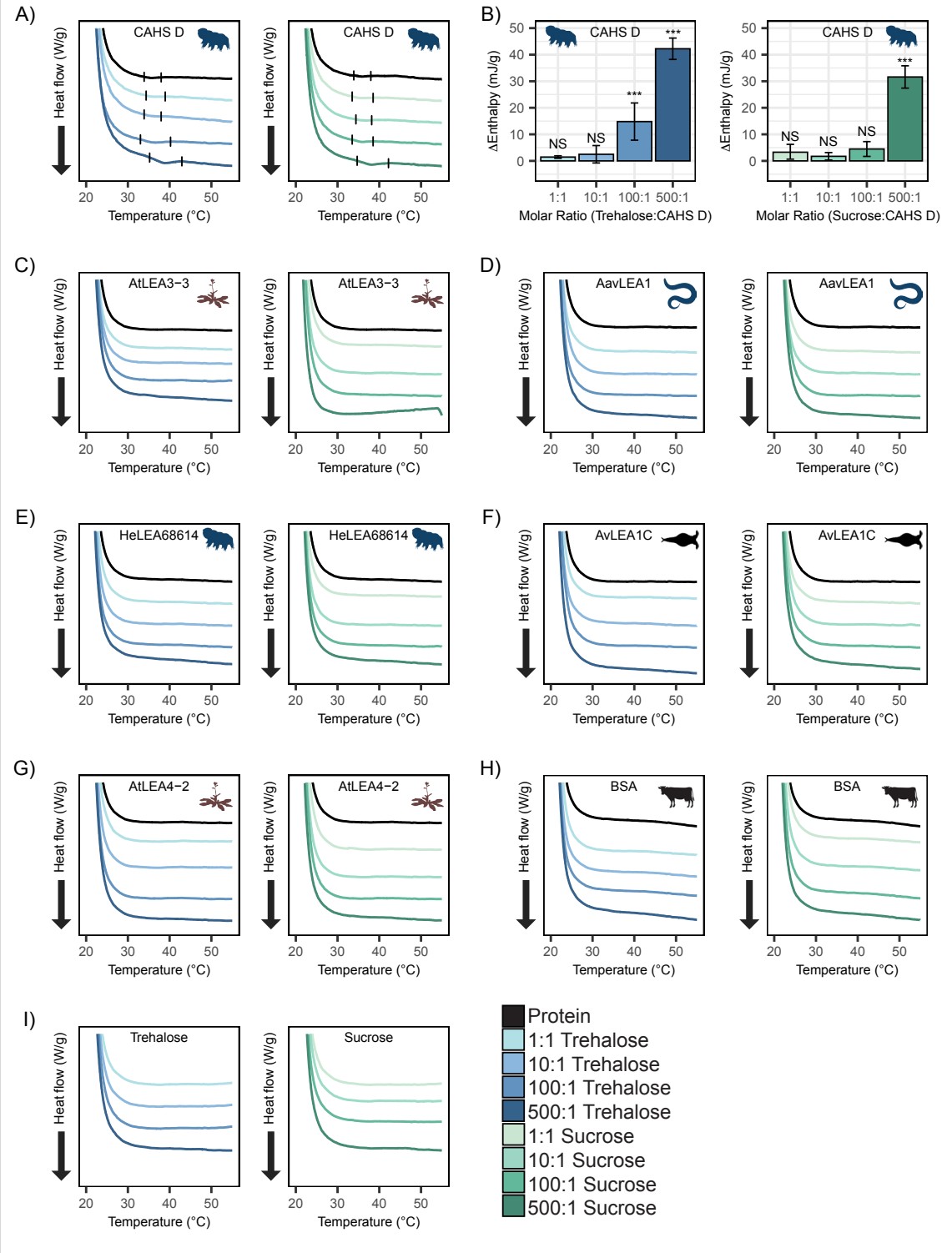

**Figure 5.** Differential scanning calorimetry (DSC) thermograms show cosolutes promote gelation of CAHS D but not late embryogenesis abundant (LEA) proteins. All DSC thermograms represent the average of three technical replicates from which the enthalpy were derived. (**A**) DSC thermogram of 0.235 mM CAHS D with trehalose (left, blue lines) and sucrose (right, green lines) at increasing cosolute:protein molar ratios. (**B**) Change in enthalpy measurements for cosolute:CAHS D mixtures relative to CAHS D. Enthalpy measurements were done by taking the area of gel melt peaks represented by black dashes in (**A**) (see 'Materials and methods'). *t*-test was used to determine pairwise comparisons between CAHS D and its cosolute mixtures. Error bars represent standard deviation. (**C–H**) DSC thermogram of LEA proteins and bovine serum albumin (BSA) at 0.235 mM and with the addition of trehalose and sucrose at molar ratios of 1:1, 10:1, 100:1, and 500:1 (cosolute:protein). (**I**) DSC thermogram for trehalose and sucrose in the absence

*Figure 5 continued on next page*

*Figure 5 continued*

of proteins at respective concentrations for different molar ratios. Organismal icons that are colored blue indicate the species uses trehalose as an endogenous cosolute, brown indicates the species uses both trehalose and sucrose, and black indicates the species uses neither trehalose nor sucrose as an endogenous cosolute.

The online version of this article includes the following source data and figure supplement(s) for figure 5:

**Source data 1.** Heat flow and enthalpy measurements for CAHS D related to *Figure 5A and B*.

**Source data 2.** Overlay of differential scanning calorimetry (DSC) thermograms for CAHS D from which the resulting enthalpy was calculated.

**Source data 3.** Heat flow and enthalpy measurements for bovine serum albumin (BSA) and late embryogenesis abundant (LEA) proteins related to *Figure 5C–I*.

**Figure supplement 1.** Differential scanning calorimetry (DSC) thermograms and calculation of enthalpy and melting points.

**Figure supplement 1—source data 1.** All data related to *Figure 5—figure supplement 1*.

(see 'Materials and methods'). Our calculations reveal that trehalose has a negative $\Delta\Delta G_{tr}^{M \to D}$ with CAHS D, meaning the dimeric state is favored in the presence of this cosolute. Sucrose's $\Delta\Delta G_{tr}^{M \to D}$ is close to 0, neither stabilizing nor destabilizing the dimer (*Figure 6B*). In addition to these cosolutes, we wanted to explore the effect of a cosolute with a positive $\Delta\Delta G_{tr}^{M \to D}$. Glycine betaine, a common stabilizing cosolute, but without known roles in tardigrade desiccation tolerance, displayed a positive $\Delta\Delta G_{tr}^{M \to D}$ (*Figure 6B*). Together these cosolutes span a range of $\Delta\Delta G_{tr}^{M \to D}$ values that are expected to increase the dimer population (trehalose), have a minimal impact on dimerization (sucrose), or increase the monomer population (glycine betaine) of CAHS D (*Auton and Bolen, 2005*; *Hong et al., 2015*). While data for trehalose and sucrose are in line with this analysis, we next sought to determine empirically the impact of glycine betaine on CAHS D gelation.

To test the effects of glycine betaine on CAHS D dimerization predicted by these TFE calculations, we first repeated our DSC and SAXS experiments in the presence of glycine betaine. Unlike trehalose and sucrose, below the protein's gelation threshold no increase in enthalpy of melting was observed upon the addition of glycine betaine (*Figure 6C and D*). To probe whether glycine betaine inhibits CAHS D oligomerization, we conducted additional DSC experiments above the protein's gelation threshold. While trehalose and sucrose enhanced gelation of CAHS D (*Figure 5—figure supplement 1B–E*), we observed a decrease in enthalpy of melting when glycine betaine was present at the 500:1 molar ratio, signifying inhibition of CAHS D gelation (*Figure 6E and F*).

We then performed SAXS on CAHS D at 4 mg/mL with 1000:1 molar ratios of each cosolute. The intent of using such a high molar ratio was to mimic the high concentration of cosolutes that CAHS D would experience during desiccation (*Romero-Perez et al., 2023*). A non-gelling concentration of CAHS D in 1000:1 glycine betaine yielded a scattering profile similar to the protein with no cosolute indicating a lack of gelation (*Figure 6G*). Meanwhile, 1000:1 sucrose and trehalose yielded scattering profiles consistent with gelation, similar to those previously reported (*Sanchez-Martinez et al., 2023*). This is made especially evident by a peak at q = 0.06 Å⁻¹, which reports on the width of CAHS D's gel fibers and matches previously reported scattering profiles for gelled CAHS D (*Sanchez-Martinez et al., 2023*). Consistent with our hypothesis that the more negative $\Delta\Delta G_{tr}^{M \to D}$ for trehalose will increase dimerization and subsequent gelation, we observed an increased curvature in the Guinier region indicating increased fibrillization (*Figure 6—figure supplement 1C and D*).

Finally, we tested the impact of glycine betaine on CAHS D's protective capacity. If induction of self-assembly of CAHS D is a mechanism underlying trehalose-/sucrose-induced synergy, then one would expect that glycine betaine's inhibition of gelation would result in no synergy or even have an antagonistic effect. Glycine betaine on its own is not protective to LDH during drying nor does it inhibit LDH activity (*Figure 6—figure supplement 1E*). We found a significant antagonistic relationship between glycine betaine and CAHS D on LDH protection (*Figure 6H*). Furthermore, Pearson correlation between the $\Delta\Delta G_{tr}^{M \to D}$ of a cosolute (scaled by concentration) and its ability to induce synergy in CAHS D was statistically significant (*Figure 6I*).

Using a similar AlphaFold2-based approach for LEA proteins and for BSA, one observes correlations between the $\Delta\Delta G$ of the disorder-to-order transition and synergy (*Figure 6—figure supplement 1F–L*). Interestingly, AlphaFold2 predictions of our LEA proteins were broadly helical, which is in contrast to our experimental characterization of these proteins in aqueous solutions (*Figure 6—figure supplement 2*). However, this is not unusual for AlphaFold2 predictions and could possibly represent

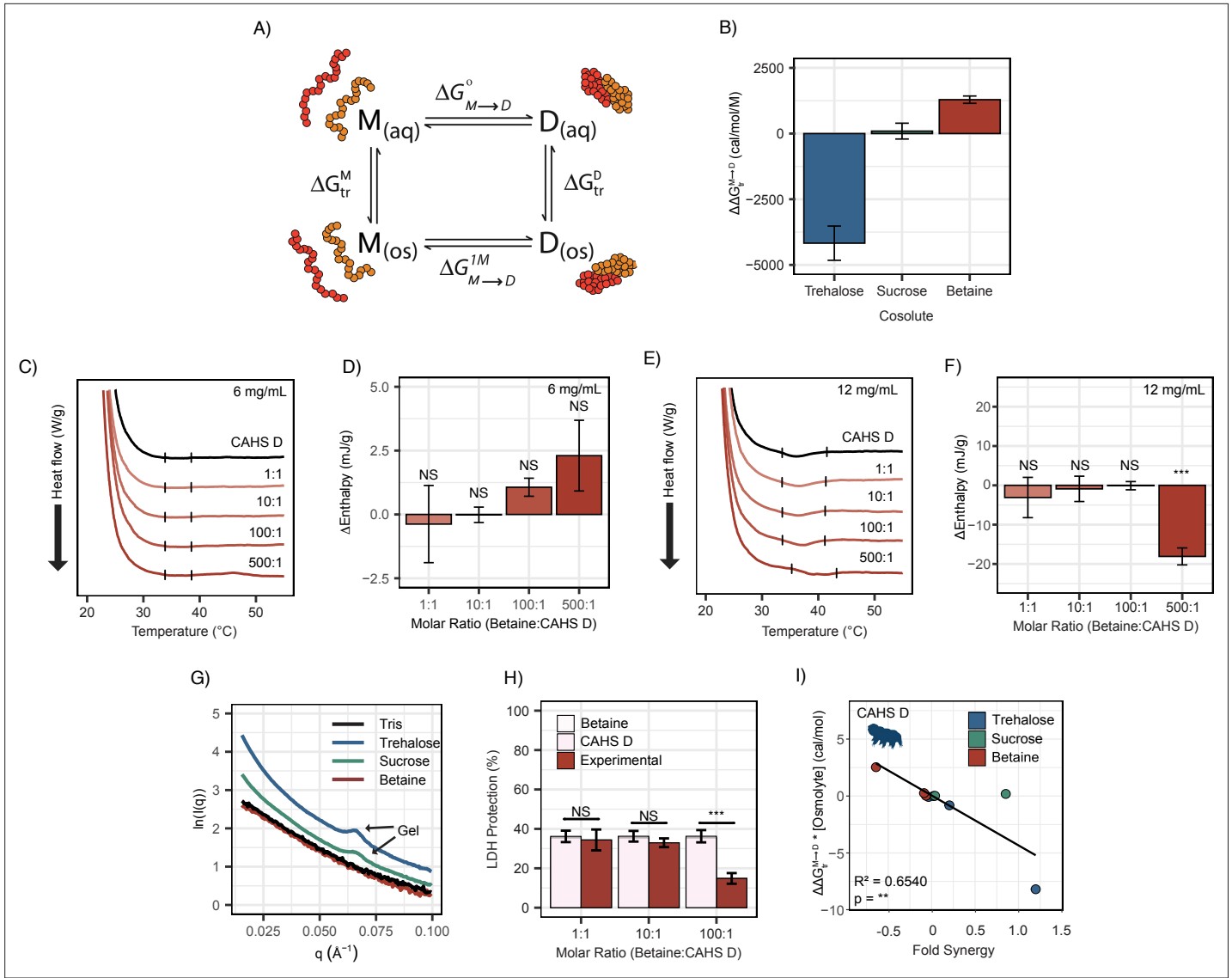

**Figure 6.** Transfer free energy of CAHS D highlights the difference between synergistic and non-synergistic cosolutes. (**A**) Tanford's transfer model, depicting the effect of cosolutes on the dimerization of two proteins. 'M'' represents the protein's monomeric state, 'D' represents the dimeric state, 'aq'' represents an aqueous solution, and 'os'' represents a solution containing some osmolyte. (**B**) The difference in $\Delta\Delta G_{tr}^{M\to D}$ between two CAHS D monomers and a CAHS D dimer. All structures are predicted by AlphaFold2 (*Jumper et al., 2021*; *Jumper et al., 2020*). (**C**) Differential scanning calorimetry (DSC) thermograms of CAHS D at 6 mg/mL (0.235 mM) in varying molar ratios of glycine betaine. (**D**) Change in enthalpy measurements for betaine:CAHS D mixtures relative to CAHS D at 6 mg/mL. Enthalpy measurements were done by taking the area of gel melt peaks represented by black dashes in (**C**). (**E**) Same as (**C**) but with CAHS D at 12 mg/mL (0.47 mM). (**F**) Change in enthalpy measurements for betaine:CAHS D mixtures relative to CAHS D at 12 mg/mL. (**G**) Small-angle X-ray scattering (SAXS) data depicting scattering profiles of 4 mg/mL CAHS D in Tris (black), 1000:1 trehalose (blue), 1000:1 sucrose (green), and 1000:1 glycine betaine (red). (**H**) Lactate dehydrogenase (LDH) synergy assay for glycine betaine and CAHS D at various molar ratios. Welch's *t*-test was used for statistical comparison. n = 3, error bars represent standard deviation. (**I**) A correlation of $\Delta\Delta G_{tr}^{M\to D}$ of a given cosolute with CAHS D and its synergy in the LDH assay at different molar ratios. p-value is given from a Pearson correlation.

The online version of this article includes the following source data and figure supplement(s) for figure 6:

**Source data 1.** All data related to *Figure 6B–I*.

**Source data 2.** Equations and overlay of differential scanning calorimetry (DSC) thermograms for CAHS D from which the resulting enthalpy was calculated.

**Figure supplement 1.** Supplementary figures for *Figure 6*.

**Figure supplement 1—source data 1.** All data related to *Figure 6—figure supplement 1*.

**Figure supplement 2.** Representative images of the pdb structures obtained from our AlphaFold2 predictions.

**Figure supplement 2—source data 1.** All data related to *Figure 6—figure supplement 2*.

a 'bound' conformation for the proteins (*Alderson et al., 2023*). For a subset of these proteins, we see a statistically significant correlation between $\Delta\Delta G$ and synergy (*Figure 6—figure supplement 1F–L*). However, this data is purely computational. For CAHS D, we saw our predictions recapitulated in changes in the protein structure, and for LEA proteins we do not. Thus, we conclude that cosolutes do not induce synergy in our LEA proteins through a change in folding.

Taken together, these results suggest that the driving force for gelation in CAHS D and its ability to synergize with a given cosolute is rooted in the direct interaction between CAHS D and the prevalent cosolute. However, trying to apply this model to the LEAs tested in this work failed to yield meaningful correlates. Thus, we propose that direct interactions between cosolute and LEA proteins cannot explain the synergy observed with cosolutes. Overall, our study demonstrates that while synergy between desiccation-related IDPs and endogenous cosolutes appears to be a widespread and conserved behavior, the mechanisms underlying this synergy vary between IDP families.

## Discussion

In this study, we examined the interplay between IDP sequence, solution environment, ensemble, and function. To do this, we have taken advantage of the dramatic changes to the cosolute content of anhydrobiotic organisms brought on by desiccation and compared the effects of these cosolutes between representatives from three families of desiccation-related IDPs (LEA_4, LEA_1, and CAHS proteins). For the proteins we tested empirically, we demonstrate that endogenous cosolutes enriched during desiccation enhance the protective capacity of CAHS D, full-length LEA_4, and LEA_1 proteins in an in vitro enzyme assay. Surprisingly, the functional changes were not accompanied by any detectable structural changes to the monomeric ensemble of these IDPs under the conditions tested. However, synergistic cosolutes did induce oligomerization and gelation of CAHS D. Finally, we show that in the case of CAHS D, but not LEAs, oligomerization, gelation, and protective synergy can be traced to direct interactions between cosolute and protective protein. Our results suggest that while functional synergy between the solution environment spans multiple IDP families, different mechanisms can underlie synergistic interactions for different proteins.

Functional synergy for the full-length proteins from different organisms mirrored the endogenous cosolute environment in that organism. In all cases, an IDP protected LDH activity more with its endogenous cosolute compared to an exogenous cosolute. These differences in synergy appear to extend across even subtle variations in cosolute use. For example, nematodes and tardigrades both accumulate trehalose during desiccation, but tardigrades accumulate orders of magnitude less (*Nguyen et al., 2022*; *Higa and Womersley, 1993*; *Browne et al., 2004*; *Hengherr et al., 2008*). Consistent with this, both tardigrade proteins used in this study synergized with trehalose at an order of magnitude lower concentration than what was required to elicit synergy with the nematode LEA protein.

It is important to note that desiccation-tolerant organisms employ multiple cosolutes to counteract the effects of desiccation. The use of a single cosolute-IDP system in our in vitro experiments does not accurately mirror the diverse cosolute changes in desiccating systems. For instance, *Arabidopsis* seeds enrich both trehalose and sucrose, among other cosolutes. This demands the necessity of future experiments that incorporate both or multiple cosolutes and assess their synergistic effects, thus elucidating the intricate synergy in multi-cosolute systems. Additionally, we want to point out that our results cannot necessarily be generalized to all desiccation-related IDPs. More experiments will be needed to assess the relevance of cosolute effects to functional synergy and IDP folding in the context of desiccation and beyond. This remains an important future direction for the field.

Overall, while our study found that synergy with endogenous cosolutes is observed across two families of LEA proteins as well as CAHS proteins, we observed a major difference in structural changes induced in these protein families. While in CAHS D synergy could be traced back to the interaction between the protective protein and the cosolute, no underlying mechanism was detected for the tested LEAs. What then could be driving the synergy we observe for these proteins?

One possibility is that in most of our assays we do not consider the protein being protected. Our studies of molecular mechanisms for synergy do not consider the underlying effect of both protectant protein and cosolute on LDH directly. It is possible that the presence of both endogenous cosolute and protein create a solvation environment that becomes highly protective, for example, rehydration. In line with this, recent studies have highlighted the ability of LEA proteins to stabilize sugar glasses in a dry state (*Shimizu et al., 2010*; *Wolkers et al., 2001*; *Shih et al., 2012*; *Hincha et al., 2021*).

Glass formation is known to preserve labile biomolecules during desiccation, contributing to survival (***Boothby et al., 2017***; ***Sakurai et al., 2008***). Different glasses vary significantly in their protective capacity, and studies have attempted to find structural properties that explain this difference (***Hincha et al., 2021***; ***Ramirez et al., 2024***). Because trehalose and sucrose both form glasses when dried (***Crowe et al., 1998***; ***Simperler et al., 2006***), it is possible that our LEA proteins are inducing a change in the glass's structural properties that leads to synergy.

Another possibility is a difference in the nature of TFE-induced oligomerization. While repulsive cosolutes drive homotypic interactions between CAHS D monomers, the same thermodynamic force can promote heterotypic interactions between LEAs and other proteins (***Riback et al., 2020***). For example, trehalose may stabilize electrostatic interactions between LEA proteins and LDH during our in vitro synergy assays. If the protective capacity of LEA proteins is dependent on direct interactions between the protectant and the client protein, then this is a plausible explanation for synergy. However, it is currently unknown whether or not this is the case.

Another major question posed by this research is why sucrose was able to elicit synergy in CAHS D. Our computational approach predicted that sucrose should be agnostic to CAHS D's ability to form dimers, and yet we clearly see that, in vitro, sucrose is a moderately potent driver of gelation. We believe several factors could explain this effect. While sucrose does not drive dimerization through direct 'soft' repulsion, it may still do so by acting as a crowder eliciting an excluded volume effect (***Sarkar et al., 2013***). Another possible manifestation of an excluded volume effect is the slight increase in the melting peak that is observed with all cosolutes used in this study at high molar ratios (***Figure 5—figure supplement 1F–I***, ***Figure 6—figure supplement 1M and N***). Alternatively, given that CAHS D must polymerize beyond the dimeric state to form a gel, sucrose may stabilize a higher-level oligomer that was not captured in our $\Delta\Delta G_{tr}^{M \to D}$ analysis.

IDPs are known to play important regulatory functions during development and disease progression that often occurs together with changes to the chemical composition of the intracellular environment. For example, there are known links between type II diabetes and Alzheimer's disease, and it has been shown that the intrinsically disordered neurodegenerative peptide Aβ42 undergoes pathological oligomerization in the presence of glucose whose levels mirror those found in diabetic patients (***Kedia et al., 2017***). Our study showcases how different cosolute environments can have a direct effect on the function of IDPs. By understanding the rules governing desiccation-related IDP-cosolute interactions, we might better understand the influence of changing chemical environments on a host of other IDPs.

# Materials and methods

**Key resources table**

| Reagent type (species) or resource | Designation | Source or reference | Identifiers | Additional information |
|---|---|---|---|---|
| Strain, strain background *Escherichia coli* BL21 (DE3) | CAHS D | This study | NA | Sequence information presented in ***Supplementary file 1*** |
| Strain, strain background *E. coli* BL21 (DE3) | AtLEA3-3 | This study | NA | Sequence information presented in ***Supplementary file 1*** |
| Strain, strain background *E. coli* BL21 (DE3) | AavLEA1 | This study | NA | Sequence information presented in ***Supplementary file 1*** |
| Strain, strain background *E. coli* BL21 (DE3) | HeLEA68614 | This study | NA | Sequence information presented in ***Supplementary file 1*** |

*Continued on next page*

*Continued*

| Reagent type (species) or resource | Designation | Source or reference | Identifiers | Additional information |
|---|---|---|---|---|
| Strain, strain background *E. coli* BL21 (DE3) | AvLEA1C | This study | NA | Sequence information presented in ***Supplementary file 1*** |
| Strain, strain background *E. coli* BL21 (DE3) | AtLEA4-2 | This study | NA | Sequence information presented in ***Supplementary file 1*** |
| Recombinant DNA reagent | CAHS D (plasmid) | This study | NA | Cloned via Gibson assembly in pEt28b |
| Recombinant DNA reagent | AtLEA3-3 (plasmid) | Twist Bioscience | NA | Cloned in pEt28a |
| Recombinant DNA reagent | AavLEA1 (plasmid) | This study | NA | Cloned via Gibson assembly in pEt28b |
| Recombinant DNA reagent | HeLEA68614 (plasmid) | This study | NA | Cloned via Gibson assembly in pEt28b |
| Recombinant DNA reagent | AvLEA1C (plasmid) | This study | NA | Cloned via Gibson assembly in pEt28b |
| Recombinant DNA reagent | AtLEA4-2 (plasmid) | This study | NA | Cloned via Gibson assembly in pEt28b |
| Peptide, recombinant protein | At11 | GenScript | NA | |
| Peptide, recombinant protein | At22 | GenScript | NA | |
| Peptide, recombinant protein | At44 | GenScript | NA | |
| Peptide, recombinant protein | Aav11 | GenScript | NA | |
| Peptide, recombinant protein | He11 | GenScript | NA | |
| Peptide, recombinant protein | Rt11 | GenScript | NA | |
| Peptide, recombinant protein | At20 | GenScript | NA | |
| Peptide, recombinant protein | BSA | Sigma-Aldrich | 9048-46-8 | |
| Peptide, recombinant protein | L-LDH | Sigma-Aldrich | 9001-60-9 | |
| Chemical compound, drug | NADH | Sigma-Aldrich | 606-68-8 | |
| Chemical compound, drug | Sodium pyruvate | TCI Chemicals | 113-24-6 | |
| Chemical compound, drug | D-Trehalose dihydrate | VWR | 6138-23-4 | |
| Chemical compound, drug | Sucrose | Sigma-Aldrich | 57-50-1 | |
| Chemical compound, drug | Betaine | Cayman Chemical | 107-43-7 | |

*Continued on next page*

*Continued*

| Reagent type (species) or resource | Designation | Source or reference | Identifiers | Additional information |
|---|---|---|---|---|
| Chemical compound, drug | (2,2'-bipyridyl)dichlororuthenium(II)hexahydrate | Sigma-Aldrich | 50525-27-4 | |
| Software, algorithm | GraphPad (version 9.5.1) | GraphPad Software | https://www.graphpad.com/ RRID:SCR_002798 | |
| Software, algorithm | R-Studio (version 4.1.2) | R-Studio | https://www.r-project.org/ RRID:SCR_001905 | |
| Software, algorithm | SOURSOP (version 0.2.4) | *Lalmansingh et al., 2023* | https://github.com/holehouse-lab/soursop | |
| Software, algorithm | AlphaFold2 (version 2.3.2) | Deepmind, *Jumper et al., 2021* | https://colab.research.google.com/github/deepmind/alphafold/blob/main/notebooks/AlphaFold.ipynb | |
| Other | Tzero aluminum pans | TA Instruments | 901683.901 | |

## Protein sequences

Sequences of all peptides and proteins used in this study are available in *Supplementary file 2*.

## Cloning

Inserts for full-length proteins AavLEA1, AtLEA4-2, AvLEA1C, CAHS D, and HeLEA68614 were synthesized as codon-optimized gBlocks (Integrated DNA Technologies) and cloned into the pET28b expression vector using Gibson assembly (New England Biosciences). AtLEA3-3 was cloned in pET28a vector by Twist Bioscience. Clones were propagated in DH5α cells (NEB, Cat# C2987H) and verified by Sanger sequencing (Eton Bioscience).

## Protein expression

Expression constructs were transformed into BL21 (DE3) cells (New England Biosciences, Cat# C2527H) and plated on Luria-Bertani (LB) agar plates with 50 µg/mL kanamycin. At least three single colonies were chosen for each construct and tested for expression. Constructs were expressed in 1 L LB/kanamycin medium and grown at 37°C while shaking at 180 rpm (Eppendorf Innova S44i) until an OD600 of 0.6 was reached. The culture was induced with 1 mM IPTG and grown for the next 4 hr while shaking. AvLEA1C was grown for 1 hr following IPTG addition. Cells were harvested by centrifugation at 4000 rpm for 30 min at 4°C. Cell pellets were resuspended in 5 mL of 20 mM Tris buffer, pH 7.5 supplemented with 30 µL of 1× protease inhibitor (Sigma-Aldrich, Cat# P2714). Cell pellets were stored at –80°C until further use.

## Protein purification

Frozen pellets were thawed at room temperature, subjected to heat lysis in boiling water for 10 min, and cooled down for 15 min. These were then centrifuged at 10,500 rpm at 10°C for 30 min, and the supernatant was later filter-sterilized through a 0.22 µm filter to remove any insoluble particles (EZFlow Syringe Filter, Cat# 388-3416-OEM). The filtrate was diluted two times the volume with buffer UA (8 M urea [Acros Organics, CAS no. 57-13-6], 50 mM sodium acetate [Tocris CAS No. 127-09-3], pH 4). This was loaded onto a HiPrep SP HP 16/10 (Cytiva, Cat# 29018183) cation exchange column and purified on an AKTA Pure (Cytiva, Cat# 29018224), controlled using the UNICORN 7-9.1 Workstation pure-BP-exp (Cytiva, Cat # 29128116). CAHS D was eluted using a 0–40% UB (8 M urea, 50 mM sodium acetate, and 1 M NaCl, pH 4) gradient and fractionated over 15 column volumes. LEA proteins were eluted using the 0–70% UB gradient over 15 column volumes. Protein fractions were assessed using SDS-PAGE and selected fractions were dialyzed in a 3.5 kDa tubing (SpectraPor 3 Dialysis Membrane, part no. 132724) in 20 mM sodium phosphate buffer pH 7, followed by six rounds of Milli-Q water (18.2 MΩcm) at 4 hr interval each. Concentrations of the dialyzed fractions were then quantified using

Qubit 4 fluorometer (Invitrogen, REF Q33226), flash frozen, then lyophilized (Labconco FreeZone 6, Cat# 7752021) for 48 hr, and stored at –20°C until further use.

## LEA motif sequence identification

LEA_4 and LEA_1 sequence motifs were identified in full-length LEA proteins using RADAR (https://www.ebi.ac.uk/Tools/pfa/radar/). In cases where RADAR was unable to identify repetitive motifs (e.g., in cases where a full-length LEA protein had only one or two motif repeats), manual selection and alignment of motifs was performed.

## LDH protection assay

LDH assay was adopted from previous studies (*Nguyen et al., 2022*; *Boothby et al., 2017*; *Goyal et al., 2005*; *Piszkiewicz et al., 2019*). Protectants were resuspended at a final concentration range 20 mg/mL to 0.01 mg/mL in 25 mM Tris-HCl pH 7. Rabbit muscle lactate dehydrogenase (LDH), sourced from Sigma (Sigma-Aldrich, Cat# 10127230001), was added to each solution at a concentration of 0.1 mg/mL. Half of this sample was dried in a vacuum desiccator (SAVANT Speed Vac Concentrator) for 16 hr, while the other half was refrigerated at 4°C for the same duration. Water was added to both desiccated and non-desiccated samples to a final volume of 250 µL each. 10 µL sample was mixed with 980 µL phosphate pyruvate buffer (100 mM sodium phosphate, 2 mM sodium pyruvate; pH 6.00) supplemented with 10 µL of 10 mM NADH (Sigma-Aldrich NADH; disodium salt, grade II) in a quartz cuvette. LDH activity was measured as the kinetics of the decrease in NADH absorption at 340 nm for a minute in NanodropOne (Thermo Scientific). Percent protection was calculated as a ratio of NADH absorbance for the desiccated samples normalized to non-desiccated controls. Each sample was performed in triplicate.

## LDH synergy assay

The protection data for individual protein or motif was used to select a suboptimal protective concentration. Trehalose or sucrose was mixed in equal parts with proteins at 2× concentration in 100 µL resuspension buffer (25 mM Tris-HCl pH 7) at respective molar ratios. LDH assay was performed for the mixtures as described previously. For each mixture, LDH protection was assessed individually and as a mixture. The sum of the protection conferred by individual protein and cosolute was determined, which would refer to the expected additive protection. Synergy was determined by statistical comparison of this expected additive protection with the experimental protection.

## Synergy calculation using equilibrium constant ($K_{app}$)

Synergy was calculated as an apparent equilibrium constant between functional and non-functional LDH. The equilibrium constant for the IDP – $K_{app(IDP)}$, cosolute – $K_{app(cosolute)}$ and the mixture – $K_{app(mixture)}$ were determined by the formula

$$K_{app} = [\text{Functional LDH}]/[\text{Non Functional LDH}]$$

The fraction for expected additive effect was calculated as

$$\text{Fraction}_{expected} = [K_{app(IDP)} + K_{app(cosolute)}]/[1 + K_{app(IDP)} + K_{app(cosolute)}]$$

The expected $K_{app}$ was then determined as

$$K_{app(expected)} = \text{Fraction}_{expected}/[1 - \text{Fraction}_{expected}]$$

Synergy was calculated as

$$\text{Synergy} = K_{app(mixture)} - K_{app(IDP)} - K_{app(cosolute)}$$

## Synergy calculation using Bliss independent model

LDH assay was used to generate the percent protection for IDP – $\text{Protection}_{IDP}$, cosolute – $\text{Protection}_{cosolute}$ and the mixture – $\text{Protection}_{mixture}$. The expected additive protection for the IDP:cosolute mixture was calculated as:

$$\text{Protection}_{\text{expected}} = \text{Protection}_{\text{IDP}} + \text{Protection}_{\text{cosolute}} - (\text{Protection}_{\text{IDP}} * \text{Protection}_{\text{cosolute}})$$

Synergy was then calculated as

$$\text{Synergy} = (\text{Protection}_{\text{mixture}} - \text{Protection}_{\text{expected}})/\text{Protection}_{\text{expected}}$$

## CD spectroscopy

CD spectroscopy was adopted from *Bremer et al., 2017*. Lyophilized proteins were resuspended in 25 mM NaPi pH 7 to a concentration of 200 μM. The resuspended protein was then mixed in equal parts with the NaPi buffer, 20 mM trehalose, or 20 mM sucrose in separate samples to a 100:1 molar ratio, with a final protein concentration of 100 μM and cosolute concentration of 10 mM. Protein concentration was confirmed with either a UV-vis (Thermo Scientific, GENESYS 50 UV-visible spectro-photometer) or a Qubit (Life Technologies, Qubit 3.0 Fluorometer). 20 μL aliquots of the samples were deposited on a 0.05 mm quartz cuvette and measured in a CD spectrometer (JASCO, J-1500 model). New 20 μL aliquots were then deposited on one half of a 0.05 mm quartz cuvette and spread across part of the cuvette with the tip of a pipette, to a surface area of about 1 cm². The samples were then desiccated in a vacuum chamber with drierite for 1 hr to create a dry film, and another CD measurement was taken immediately after the vacuum was stopped. Each measurement was performed in triplicate.

## SAXS: Sample preparation

Lyophilized protein was resuspended at high concentration in a buffer containing 20 mM Tris-HCl (pH 7.0) and the correct amount of cosolute to reach the desired molar ratio. Protein samples were then quantified with the Qubit Protein Assay from ThermoFisher Scientific (Cat# Q33212). The proteins were then diluted into 8 mg/mL and 4 mg/mL stocks using the same cosolute solution. Due to the necessity for each sample to have an identical buffer blank, the concentration of cosolute in the 8 mg/mL sample had to be the same as in the 4 mg/mL sample, meaning the molar ratio would be doubled in the 4 mg/mL sample. For each sample, a small aliquot of buffer was saved and stored at 4°C for use as a blank. Samples and buffer blanks were filtered using 0.22 μm syringe filters and loaded into an Axygen 96-well polypropylene PCR Microplate (Corning product# PCR-96-FS-C), which was then sealed with an AxyMat Sealing Mat (product# AM-96-PCR-RD) and wrapped in parafilm. Plates were shipped to Lawrence Berkeley National Labs in a styrofoam cooler filled with cold packs. All SAXS measurements were performed by the SIBYLS group at the Lawrence Berkeley National Laboratory HT-SAXS beamline (12.3.1) (*Dyer et al., 2014*; *Trame et al., 2004*). For technical restrictions, proteins were measured at 20:1 and 200:1 molar ratios of trehalose and sucrose instead of 10:1 and 100:1. The exception to this was CAHS D, which was tested with a wider range of molar ratios (1.6:1, 16:1, and 160:1).

We note that the radius of gyration that we calculated for CAHS D is approximately 5 angstroms greater than previously reported by our group (*Sanchez-Martinez et al., 2023*). We believe that this difference can be attributed to minor differences in our approach. While we used size-exclusion chromatography (SEC)-coupled SAXS and a relatively dilute CAHS D sample in our previous study, the number of SAXS experiments in this study necessitated a higher-throughput approach that omitted the SEC step. We therefore believe that our control samples contained a higher fraction of transient oligomeric species, which inflated the radius of gyration without significantly curving the Guinier region. Given that this fact was consistent between all of our CAHS D samples, the inter-environmental comparisons are still valid.

## SAXS: Guinier and pMW analysis

Notable aggregation, likely induced by exposure to X-rays, was present in some samples, especially in solutions that contained cosolutes. This was controlled for by excluding scattering data from samples that had already been exposed to large amounts of X-ray radiation and were thus statistically different from the initial readings. Despite some readings having been excluded, a Guinier analysis was able to be conducted for each protein:cosolute combination. Buffer subtractions and Guinier analysis were performed using BioXTAS RAW v. 2.1.4 (*Nielsen et al., 2009*; *Hopkins et al., 2018*). A qMaxR$_g$ of 1.1 was used to establish linear fits in the Guinier region (*Kachala et al., 2015*; *Rieloff and Skepö, 2020*). Samples with Guinier regions that could not be fit were excluded from the study. Molecular

weight approximations were performed using the method described in *Hajizadeh et al., 2018*, which is programmed directly into BioXTAS RAW (*Hajizadeh et al., 2018*). The 8 mg/mL samples tended to be far more aggregation-prone than the 4 mg/mL samples, so only the 4 mg/mL data is reported here.

## PICUP

PICUP was performed as previously described (*Rivera-Najera et al., 2014*; *Liu et al., 2017*). Briefly, lyophilized protein, Ru(II)bpy32+, and ammonium persulfate were resuspended in 20 mM Tris pH 7.5. Each reaction mixture constituted protein at the desired concentration with 1.25 mM Tris (2,2'-bipyridyl)dichlororuthenium(II)hexahydrate (Sigma, CAS no. 50525-27-4) and 2.5 mM ammonium persulfate (Sigma, CAS no. 7727-54-0) to a final volume of 10 µL. For mixtures, cosolutes and proteins were mixed at a 100:1 molar ratio at 2× molar concentration. Photoreaction was triggered by flashing 72 W light through a 2.5 cm water filter for 10 s in a dark room. The reaction was immediately quenched by adding 10 µL of 2× Laemmli buffer containing 4% SDS and 10% β-mercaptoethanol. The reaction mixture was heated at 95°C for 5 min. 8.5 µL of each sample was run in denaturing SDS-PAGE gels and stained with Coomassie blue to visualize the oligomeric states.

## DSC measurements

Samples were prepared in Eppendorf tubes at the desired molar ratios with cosolutes. Protein mixtures were resuspended and incubated at 55°C for 5 min to ensure proper solubility. 25 µL of the sample was hermetically sealed into a previously massed pair of DSC aluminum hermetic pan and hermetic lid (Cat# 900793.901 and 901684.901, respectively, TA Instruments). The sample mass was determined after the sample was sealed within the pan and lid. The sealed samples were then run on a TA DSC2500 instrument. The DSC method for heating experiments is as follows.

Samples were equilibrated at 20°C, heated to 60°C at a 5°C per minute ramp, and then cooled to 20°C at a 5°C per minute ramp. Samples were held for a 10 min isothermal hold at 20°C and heated to 60°C at a 5°C per minute ramp. TRIOS software (TRIOS version #5.0.0.44608, TA Instruments) was used to analyze enthalpy for samples showing the melt curves. The changes in enthalpy for the mixtures were calculated relative to the protein.

## AlphaFold2 structural modeling

Protein structure predictions (both monomeric and multimeric) were performed using Google's AlphaFold2 Colab notebook. The setting 'relax_use_gpu' was checked to increase the speed of individual predictions. The number of recycles was left at 3 (*Jumper et al., 2021*). Representative images of each structure are provided (*Figure 6—figure supplement 2*). All analyses were done in triplicate to account for variation in AlphaFold2's predictions. All pdb files can be found in supplementary data.

## TFE calculations

TFE values of each cosolute for each amino acid were pulled from existing literature (*Auton and Bolen, 2005*; *Hong et al., 2015*; *Auton and Bolen, 2004*). We used experimentally derived TFE values for the transfer of a chemical group – amino acid side chains or backbone – into 1 M solutions of trehalose, sucrose, or glycine betaine (*Auton and Bolen, 2007*; *Auton and Bolen, 2005*; *Hong et al., 2015*). We then calculated the TFE for each conformation using the formula

$$\Delta G_{tr} = \sum_N \sum_{i=1} \alpha_i g_N$$

Here, $\Delta G_{tr}$ is the TFE of a protein conformation from water to 1 M cosolute solution, $N$ is the chemical group, $i$ is a numerical index for all instances of the chemical group, $\alpha$ is the surface area of the specific instance of the chemical group in square angstrom, and $g$ is the experimental value of the TFE for that chemical group per square angstrom of exposed surface area (*Auton and Bolen, 2007*). By doing this for two conformations of a protein, one can find the change in free energy of conformational change that can be attributed to the presence of an osmolyte.

Importantly, two 'end-state' protein conformations are required to calculate a $\Delta\Delta G_{tr}$ value, which is typically done for well-folded proteins (*Auton and Bolen, 2007*). However, we believe the proteins used in this study constitute an important exception. For CAHS, several different groups have noted the propensity of CAHS proteins for oligomerization, and recent research has identified the dimer as

a particularly stable CAHS D conformer (*Sanchez-Martinez et al., 2023*; *Malki et al., 2022*; *Tanaka et al., 2022*). This notion is supported by the crosslinking data, in which the CAHS D dimer is especially prominent (*Sanchez-Martinez et al., 2023*). We thus use the following equation for the effects of cosolutes on CAHS dimerization:

$$\Delta\Delta G_{tr}^{M\to D} = \Delta G_{tr}^{D} - 2 \times \Delta G_{tr}^{M}$$

where $\Delta G_{tr}^{M}$ and $\Delta G_{tr}^{D}$ is the TFE for the monomeric and dimeric state, respectively. The monomer and dimer conformations were obtained from AlphaFold2 Multimer and AlphaFold2 predictions, respectively (see AlphaFold2 method). Surface area for residues was calculated using SOURSOP, a Python package for protein structure analysis (*Lalmansingh et al., 2023*).

To calculate the $\Delta\Delta G_{tr}^{folding}$ for our LEA proteins and for BSA, we compared an AlphaFold2 prediction of a monomeric protein with a theoretical conformation in which all residues are 100% exposed. This is because (a) LEAs showed no tendency to form a gel and (b) the CD spectra switched between a primarily disordered conformation to a helical conformation upon desiccation (*Figure 3*). For all LEAs, we calculated

$$\Delta\Delta G_{tr}^{folding} = \Delta G_{tr}^{folding} - \Delta G_{tr}^{initial}$$

Here, $\Delta G_{tr}^{initial}$ represents the free energy of a completely denatured protein chain where the maximum theoretical accessibility is achieved (RASA = 1). $\Delta G_{tr}^{initial}$ represents the free energy of the protein's 'native' conformation (as determined by an AlphaFold2 prediction). AlphaFold2 predictions of our LEA proteins were broadly helical (*Figure 6—figure supplement 2*), and thus were used to represent the disorder-to-helix transition commonly observed in LEA proteins.

## Data analysis and visualization
LDH protection was fitted into a sigmoidal curve by fitting a 5PL regression analysis using GraphPad Prism v9.5.1 from which the resulting PD50 values were derived. Other plots were plotted using R-Studio. Annotation for statistical significance includes NS, $p > 0.05$, $*p \leq 0.05$, $**p \leq 0.01$, $***p \leq 0.001$.

## Acknowledgements

Support for this project came from NSF via the IntBio research program under awards 2128069 to TCB, 2128067 to SS, and 2128068 to ASH. SK and KN were supported in part by the USDA National Institute of Food and Agriculture, Hatch project #1012152. In addition, this work was made possible in part through support from an Institutional Development Award (IDeA) from the National Institute of General Medical Sciences of the National Institutes of Health (Grant # 2P20GM103432). We thank members of the Water and Life Interface Institute (WALII), supported by NSF DBI grant #2213983, for helpful discussions. We thank Dr. Greg Hura and Kathryn Burnett for their correspondence and help in performing the SAXS experiments. SAXS experiments were conducted at the Advanced Light Source (ALS), operated by Lawrence Berkeley National Laboratory on behalf of the Department of Energy, Office of Basic Energy Sciences, through the Integrated Diffraction Analysis Technologies (IDAT) program, supported by DOE Office of Biological and Environmental Research.

## Additional information

### Competing interests
Alex S Holehouse: Scientific consultant with Dewpoint Therapeutics and is on the Scientific Advisory Board of Prose Foods. The work reported here was not influenced by these affiliations. The other authors declare that no competing interests exist.

## Funding

| Funder | Grant reference number | Author |
| --- | --- | --- |
| National Science Foundation | 2128069 | Thomas C Boothby |
| National Science Foundation | 2128067 | Shahar Sukenik |
| National Science Foundation | 2128068 | Alex S Holehouse |
| USDA National Institute of Food and Agriculture | 1012152 | Kenny H Nguyen Shraddha KC |
| National Institute of General Medical Sciences | 2P20GM103432 | Thomas C Boothby |

The funders had no role in study design, data collection and interpretation, or the decision to submit the work for publication.

## Author contributions

Shraddha KC, Kenny H Nguyen, Conceptualization, Data curation, Formal analysis, Investigation, Visualization, Methodology, Writing – original draft, Writing – review and editing; Vincent Nicholson, Data curation, Formal analysis, Investigation, Visualization, Methodology, Writing – original draft, Writing – review and editing; Annie Walgren, Investigation, Writing – review and editing; Tony Trent, Software, Writing – review and editing; Edith Gollub, Data curation, Formal analysis, Investigation, Methodology, Writing – review and editing; Paulette Sofia Romero-Perez, Formal analysis, Investigation, Methodology, Writing – review and editing; Alex S Holehouse, Conceptualization, Supervision, Funding acquisition, Writing – review and editing; Shahar Sukenik, Conceptualization, Supervision, Funding acquisition, Methodology, Writing – review and editing; Thomas C Boothby, Conceptualization, Supervision, Funding acquisition, Methodology, Writing – original draft, Writing – review and editing

## Author ORCIDs

Shraddha KC ● https://orcid.org/0000-0003-3317-988X
Alex S Holehouse ● https://orcid.org/0000-0002-4155-5729
Shahar Sukenik ● https://orcid.org/0000-0003-3855-9574
Thomas C Boothby ● https://orcid.org/0000-0002-8807-3268

Reviewer #1 (Public Review): https://doi.org/10.7554/eLife.97231.3.sa1
Reviewer #2 (Public Review): https://doi.org/10.7554/eLife.97231.3.sa2
Author response https://doi.org/10.7554/eLife.97231.3.sa3

---

# Additional files

## Supplementary files

• Supplementary file 1. Concentration for synergy assays derived from the concentration range of LDH assays.
• Supplementary file 2. Sequences of all peptides and proteins.
• MDAR checklist
• Source code 1. All codes used in this study.

## Data availability

All data generated or analyzed during this study are included in this published article. All custom code used in this study is included in this published article (*Source code 1*).

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
