## [Editor Report · eLife assessment]

This **important** study investigates the sensitivity to endogenous cosolvents of three families of intrinsically disordered proteins involved with desiccation. The findings, drawn from well-designed experiments and calculations, suggest a functional synergy between sensitivity to small molecule solutes and convergent desiccation protection strategy. The evidence is found to be **convincing**, and the authors provide appropriate caveats since the study's conclusions are based on a small number of proteins. This work will be of interest to biochemists and biophysicists interested in the conformation–function relationship of intrinsically disordered proteins.

---

## [Referee Report · Reviewer #1 (Public Review)]

The individual roles of both cosolvents and intrinsically disordered proteins (IDPs) in desiccation have been well established, but few studies have tried to elucidate how these two factors may contribute synergistically. The authors quantify the synergy for the model and true IDPs involved with desiccation and find that only the true IDPs have strong desiccation tolerance and synergy with cosolvents. Using these as model systems, they quantify the local (secondary structure vis-a-vi CD spectroscopy) and global dimensions (vis-a-vi the Rg of SAXS experiments) and find no obvious changes with the co-solvents. Instead, they focus on the gelation of one of the IDPs and, using theory and experiments, suggest that the co-solvents may enable desiccation tolerance, an interesting hypothesis to guide future in vivo desiccation studies. A few minor points that remained unclear to this reviewer and that were noted previously have been reasonably addressed in this revision.

Strengths:

This paper is quite extensive and has significant strengths worth highlighting. Notably, the number and type of methods employed to study IDPs are quite unusual, employing CD spectroscopy, SAXS measurements, and DSC. The use of the TFE is an exciting integration of the physical chemistry of cosolvents into the desiccation field is a nice approach and a clever way of addressing the gap of the lack of conformational changes depending on the cosolvents. Furthermore, I think this is a major point and strength of the paper; the underlying synergy of cosolvents and IDPs may lie in the thermodynamics of the dehydration process.

Figure S6A is very useful. I encourage readers who are confused about the DSC analysis, interpretation, and calculation to refer to it.

Weaknesses:

All minor weaknesses were addressed in this revision.

---

## [Referee Report · Reviewer #2 (Public Review)]

Summary:

The paper aims to investigate the synergies between desiccation chaperones and small molecule cosolutes, and describe its mechanistic basis. The paper reports that IDP chaperones have stronger synergies with the cosolutes they coexist with, and in one case suggests that this is related to oligomerization propensity of the IDP.

Strengths:

The authors have done a good job improving the paper. The study uses a lot of orthogonal methods and the experiments are technically well done. They are addressing a new question that has not really been addressed previously.

Weaknesses:

The conclusions are still based on a few examples and only partial correlations. However, this is now acknowledged by the authors and the conclusions are presented with appropriate caveats.

---

## [Author Response]

The following is the authors’ response to the original reviews.

**Public Reviews:**

**Reviewer #1 (Public Review):**
Summary:The individual roles of both cosolvents and intrinsically disordered proteins (IDPs) in desiccation have been well established, but few studies have tried to elucidate how these two factors may contribute synergistically. The authors quantify the synergy for the model and true IDPs involved with desiccation and find that only the true IDPs have strong desiccation tolerance and synergy with cosolvents. Using these as model systems, they quantify the local (secondary structure vis-a-vi CD spectroscopy) and global dimensions (vis-a-vi the Rg of SAXS experiments) and find no obvious changes with the co-solvents. Instead, they focus on the gelation of one of the IDPs and, using theory and experiments, suggest that the co-solvents may enable desiccation tolerance, an interesting hypothesis to guide future in vivo desiccation studies. A few minor points that remain unclear to this reviewer are noted.Strengths:This paper is quite extensive and has significant strengths worth highlighting. Notably, the number and type of methods employed to study IDPs are quite unusual, employing CD spectroscopy, SAXS measurements, and DSC. The use of the TFE is an exciting integration of the physical chemistry of cosolvents into the desiccation field is a nice approach and a clever way of addressing the gap of the lack of conformational changes depending on the cosolvents. Furthermore, I think this is a major point and strength of the paper; the underlying synergy of cosolvents and IDPs may lie in the thermodynamics of the dehydration process.Figure S6A is very useful. I encourage readers who are confused about the DSC analysis, interpretation, and calculation to refer to it.Weaknesses:Overall, the paper is sound and employs strong experimental design and analysis. However, I wish to point out a few minor weaknesses.Perhaps the largest, in terms of reader comprehension, focuses on the transition between the model peptides and real IDPs in Figures 1 and 2. Notably, little is discussed with respect to the structure of the IDPs and what is known. Notably, I was confused to find out when looking at Table 1 that many of the IDPs are predicted to be largely unordered, which seemed to contrast with some of the CD spectroscopy data. I wonder if the disorder plots are misleading for readers. Can the authors comment more on this confusion? What are these IDPs structurally?

We apologize for the confusion caused here and thank the reviewer for this astute observation. Our CD spectroscopy data suggests all LEA proteins are almost entirely disordered under aqueous conditions, with a single major minimum at 200 nm, although most have a small inflection around 220 nm, indicating a small proportion of helicity (Figure 3A). The notable exception here is CAHS D, which – in line with our work and the work of many others – possesses a substantial degree of transient helicity in the linker region (residues 100-200), giving rise to a more pronounced minimum at 220 nm. These conclusions are consistent with our SAXS data (Figure 4), which predict a radius of gyration far larger than a globular folded protein of the same number of residues should have (15-20 Å). The structural predictions (both Metapredict and AlphaFold2), however, imply several of the proteins to be ordered; AvLEA1C and HeLEA68614 are both predicted to have large folded regions based on metapredict disorder scores. We believe this is an erroneous prediction driven by these regions' propensity to acquire helicity in the context of desiccation (Fig 3B) and/or when interacting with clients. As such, our computational analysis is at odds with the experimental data because these proteins are all poised to undergo a coil-to-helix transition, an effect our parallel work has proposed is important for their function (see Biswas et al. Prot. Sci. 2024). The ability of AlphaFold2 to predict bound-state or transient helices has been previously documented (Alderson et al PNAS 2023)

To address this discrepancy, the caption for Table 1 reads: “We note that the reason many of these profiles contain large folded regions is because the amphipathic LEA and CAHS proteins are predicted to form helices, which metapredict infers and incorrectly highlights these regions as ‘folded’ when really they are disordered in isolation”. We have also added additional context and information to the caption for Figure 6—figure supplement 2 “We note that the structural predictions from AlphaFold2 contain largely ordered structures. We believe this is due to the propensity of these proteins to form helices in the context of drying or when interacting with a client. This has been shown in cases where an IDR contains residual helicity or is folded upon binding [70].”

Related to the above thoughts, the alpha fold structures for the LEA proteins are predicted (unconfidently) as being alpha-helical in contrast to the CD data. Does this complicate the TFE studies and eliminate the correlation for the LEA proteins?

AlphaFold2 predicted helicity in disordered regions is commonly observed, and thought to indicate a possible “bound” helical state (Alderson et al. PNAS 2023). As shown by the CD data, in aqueous conditions no secondary structure exists. It is only in the desiccated state - the path to which requires proteins to reach excessively high concentrations - that this secondary structure appears. Underlying our TFE model is that the AlphaFold2 predicted secondary structure is indicative of the state the proteins are in at high abundance, which occurs as cells ramp up protectant expression and as water is removed from the system. Under these assumptions, the CD data is in agreement with the AlphaFold2 predictions, and our analysis holds. This is explained in the methods under “Transfer Free Energy (TFE) Calculations” - but we have now also added an additional sentence to this effect in the main text: “Using a similar AlphaFold2-based approach for LEA proteins and for BSA, we can make correlations between the Gtr of the disorder-to-order transition and synergy (Figure 6—figure supplement 1F–K). Interestingly, AlphaFold2 predictions of our LEA proteins were broadly helical, which is in contrast to our experimental characterization of these proteins in aqueous solutions. However, this is not unusual for AlphaFold2 predictions and could possibly represent a “bound” conformation for the proteins [70].”

Additionally, the notation that the LEA and BSA proteins do not correlate is unclear to this reviewer, aren't many of the correlations significant, having both a large R^2 and significant p-value?

We thank the reviewer for pointing this out. While BSA and some LEA proteins have ΔΔGtrfolding values that correlate with synergy, there’s more to consider in assessing the relevance of these correlations. For example, we cannot claim that the ΔΔGtrfolding value is physiologically relevant without observing an actual structural change in the protein. Furthermore, several of these proteins (BSA and AvLEA1C) were found to be not significantly synergistic in the LDH assay, and any correlation should, therefore, also be considered non-significant. We have added a sentence to the results to clarify this: “For a subset of these proteins, we see a statistically significant correlation between G and synergy. However, this data is purely computational. For CAHS D, we saw our predictions recapitulated in changes in the protein structure, and for the LEA proteins we do not. Thus, we conclude that cosolutes do not induce synergy in our LEA proteins through a change in folding.”

The calculation of synergy seems too simplistic or even problematic to me. While I am not familiar with the standards in the desiccation field, I think the approach as presented may be problematic due to the potential for higher initial values of protection to have lower synergies (two 50%s for example, could not yield higher than 100%).

We acknowledge the reviewer’s concern about our synergy calculation. We would like to highlight the use of sub-optimal protective concentrations in our synergy assays similar to studies previously reported in the desiccation field (Nguyen et al. 2022; Kim et al. 2018).

As the reviewer pointed out, we agree that there is a theoretical 100% threshold in our experiments which if we hit, we cannot distinguish between individual additive vs synergistic effects. To avoid the situation of reaching the near maximal protection levels (~100%), we intentionally select a sub-optimal concentration of the protectants that are below the maximum efficacy level for individual protectants to use in our assays. This limits the potential for initial higher values of the protectants so that their combined effect is not maximized, and there is always the potential for synergy. We would also like to point out that we never actually hit that 100% threshold in any of our synergy experiments, which warrants that any observed increase in protection is attributed to a true synergistic effect between the protectants.

Instead, I would think one would need to really think of it as an apparent equilibrium constant between functional and non-functional LDH (Kapp = [Func]/[Not Func] and frac = Kapp/(1+Kapp) or Kapp = frac/(1-frac)). Then after getting the apparent equilibrium constants for the IDP and cosolvent (KappIDP and KappCS), the expected additive effect would be frac = (KappIDP+KappCS)/(1+KappIDP+KappCS).Consequently, the extent of synergy could be instead calculated as KappBOTH-KappIDP-KappCS. Maybe this reviewer is misunderstanding. It is recommended that the authors clarify why the synergy calculation in the manuscript is reasonable.

We thank the reviewer for this suggestion. In the desiccation field, the synergy calculations that we used is the standard method that people use, so that’s what we present in our main manuscript. However, we have now quantified synergy through two new approaches: one, as suggested by the reviewer, using the equilibrium constant (Kapp) as a metric, and the other using the Bliss Independent model, which is a common approach for calculating synergy in drug combination studies. We see minimal differences in terms of the synergy scores using these different methods. We have included the results for these additional methods in supplemental figure S3.

Related to the above, the authors should discuss the utility of using molar concentration instead of volume fraction or mass concentration. Notably, when trehalose is used in concentration, the volume fraction of trehalose is much smaller compared to the IDPs used in Figure 2 or some in Figure 1. Would switching to a different weighted unit impact the results of the study, or is it robust to such (potentially) arbitrary units?

We thank the reviewer for this comment. Indeed, in studies of cosolute effect, concentration units can alter the conclusions of the study (Auton and Bolen 2004). In our case, the relevant figures where we use a concentration scale (1B and 2B) are not germane to the main conclusions: The only use of these PD50 values is to determine a sub-optimal concentration at which ~30% of the LDH is protected. While it is true that the number for the concentration of e.g., trehalose will be dramatically different if we were to use mass fraction units, the rest of the work and all our conclusions would be exactly the same.

Additionally, our use of a molar ratio when discussing synergy is a direct result of the way we think about such synergy: Since the concentration of both protein and cosolute can change by orders of magnitude during drying, it is the copy numbers of both proteins and cosolute that are conserved in this process, and it is this unit that we think is important to the protective effect (rather than the partial molar volume, for example, which would be changing as the system dries).

**Reviewer #2 (Public Review):**
Summary:The paper aims to investigate the synergies between desiccation chaperones and small molecule cosolutes, and describe its mechanistic basis. The paper reports that IDP chaperones have stronger synergies with the cosolutes they coexist with, and in one case suggests that this is related to oligomerization propensity of the IDP.Strengths:The study uses a lot of orthogonal methods and the experiments are technically well done. They are addressing a new question that has not really been addressed previously.Weaknesses:The conclusions are based on a few examples and only partial correlations. While the data support mechanistic conclusions about the individual proteins studied, it is not clear that the conclusions can be generalized to the extent proposed by the authors due to small effect sizes, small numbers of proteins, and only partial correlations.

Thank you for bringing this up. We agree that we should not generalize our results to other systems based on the evidence we have for the proteins used in our study. We have altered our discussion to highlight that this may apply to other IDPs, and that future experiments must be done to support this: “Additionally, we want to point out that our results cannot necessarily be generalized to all desiccation-related IDPs. More experiments will be needed to assess the relevance of cosolute effects to functional synergy and IDP folding in the context of desiccation and beyond. This remains an important future direction for the field.”

The authors pose relevant questions and try to answer them through a systematic series of experiments that are all technically well-conducted. The data points are generally interpreted appropriately in isolation, however, I am a little concerned about a tendency to over-generalize their findings. Many of the experiments give negative or non-conclusive results (not a problem in itself), which means that the overall storyline is often based on single examples.

We agree with the reviewer’s point. As mentioned earlier, we have modified our manuscript to reflect that our findings are based on the six proteins that we studied, and we can only speculate about other desiccation-related IDPs based on our results.

For example, the central conclusion that IDPs interact synergistically with their endogenous co-solute (Figure 2E) is largely driven by one outlier from Arabidopsis. The rest are relatively close to the diagonal, and one could equally well suggest that the cosolutes affect the IDPs equally (which is also the conclusion in 1F).

We appreciate the reviewer’s concern regarding our conclusion in Figures 2E and 1F. We would like to highlight that our conclusions that IDPs interact synergistically with their endogenous cosolute are based on statistical analysis. Our data shows that full-length proteins that were synergistic with both cosolutes are always significantly more synergistic with the endogenous cosolute (Figure 2E, Figure 2—figure supplement 1C–E). For example, the nematode protein is synergistic with both trehalose and sucrose, but is significantly more synergistic with trehalose, the endogenous nematode cosolute, than with sucrose (Fig S2D).

This is not the case in 1F. In Figure 1F, it is to note that not only are the points close to the diagonal, but most points are close to zero along both axes indicating no synergy. In fact, many points have negative synergy (antagonistic effect).

We do recognize that our conclusions are based on the study of a specific set of six IDPs, and we do not want to overreach in our conclusions. To acknowledge this, we have now added text to emphasize that our conclusion is based on the six proteins that we tested, and we speculate it might apply to other systems: “Our data shows that these six IDPs synergize best with their endogenous cosolute to promote desiccation tolerance and we speculate that this may apply to other desiccation-related IDPs”.

Similarly, the mechanistic explanations tend to be based on single examples. This is somewhat unavoidable as biophysical studies cannot be done on thousands of proteins, but the text should be toned down to reflect the strength of the conclusions.

We acknowledge the reviewer’s concern. We have modified our manuscript accordingly to reflect that the mechanistic insights we gained are for the six proteins we tested empirically. These changes can be found throughout the manuscript. None of our experiments rule out the possibility that other LEA proteins or CAHS proteins may show different structural transitions, or that other IDPs may take on structural changes in response to the cosolutes.

The central hypothesis revolves around the interplay between cosolutes and IDP chaperones comparing chaperones from species with different complements of cosolutes. In Table 1, it is mentioned that Arabidopsis uses both trehalose and sucrose as a cosolute, yet experiments are only done with either of these cosolutes and Arabidopsis is counted in the sucrose column. While it makes sense to compare them separately from a biophysical point of view, the ability to test the co-evolution of these systems is somewhat diminished by this. At least it should be discussed clearly.

We appreciate the reviewer’s comment. As is mentioned in Table 1, *Arabidopsis* uses both trehalose and sucrose as cosolute. As such, we would predict that the *Arabidopsis* proteins would respond positively to both cosolutes. We would like to point out that *Arabidopsis* is counted in both trehalose and sucrose columns.

We would also like to emphasize that multiple osmolytes exist in all organisms as a desiccation response and a simple IDP-cosolute system is far from a true recapitulation of a desiccating system. We have touched on this in the discussion and explicitly addressed the presence of both cosolutes in *Arabidopsis* and the need for further experiments to test for synergistic interactions using both or multiple mediators to illustrate synergy in multiple cosolute systems: “It is important to note that desiccation-tolerant organisms employ multiple cosolutes to counteract the effects of desiccation. The use of a single cosolute-IDP system in our in vitro experiments does not accurately mirror the diverse cosolute changes in desiccating systems. For instance, *Arabidopsis* seeds enrich both trehalose and sucrose, among other cosolutes. This demands the necessity of future experiments that incorporate both or multiple cosolutes and assess their synergistic effects, thus elucidating the intricate synergy in multi-cosolute systems.”

It would be helpful if the authors could spell out the theoretical basis of how they quantify synergy. I understand what they are doing - and maybe there are no better ways to do it - but it seems like an approach with limitations. The authors identify one in that the calculation only works far from 100%, but to me, it seems there would be an equally strict requirement to be significantly above 0%. This would suggest that it is used wrongly in Figure 6H, where there is no effect of betaine (at least as far as the color scheme allows one to distinguish the different bars). In this case, the authors cannot really conclude synergy or not, it could be a straight non-synergistic inhibition by betaine.

We appreciate the reviewer’s concern about the theoretical basis of how we quantify synergy. We do acknowledge the limitation of our LDH protection/synergy assay only produces interpretable data when our protectant/mixture yields protection levels within the range 0 and below 100%. Betaine was not protective in any of the concentrations we tested in this study. In line with the reviewer’s comment, we also acknowledge that within our experimental procedures, the inhibitory effects of betaine cannot be accurately captured, considering that LDH activity is ~0% without protectants. However, in our positive control in which LDH is co-incubated with betaine or betaine and CAHS D overnight in the hydrated state, we do not see a loss of enzymatic function of LDH nullifying a direct inhibition by betaine. We have added this text in our manuscript: “Glycine betaine on its own is not protective to LDH during drying nor does it inhibit LDH activity (Figure 6—figure supplement 1E)”.

**Recommendations for the authors:**

**Reviewer #1 (Recommendations For The Authors):**
The conclusion in lines 195-196 seems overstated as the length dependence could be strongly changed in non-tested concentrations or those that are not possible experimentally. Notably, the IDPs in Figure 2 are around 200AA and only transition in the ranges tested for these peptides. Some other conclusions around this point seem a little overstated.

We acknowledge the reviewer’s concern about the potential variability of the length dependence of the motifs at concentrations beyond those tested. However, we would like to highlight that higher concentrations of the tandem repeats (At22 and At44) inactivated LDH during the incubation period, as was seen with the 11-mer motifs. This meant we could not evaluate protection by these motifs at concentrations beyond those plotted in Figure 1A. This behavior was not observed for the full-length proteins. Regardless, we have toned down the conclusion in lines 195-196 to only reflect our results for the 2X and 4X repeats of At11 which now reads “We synthesized 2X (At22) and 4X (At44) tandem repeats of the *A. thaliana* 11-mer LEA_4 motif (At11). At22 and At44 show minimal potency in preserving in vitro LDH function during drying (Figure 1A, Figure 1—figure supplement 1A).”

**Reviewer #2 (Recommendations For The Authors):**
Figure 3: The focus on the ratio 222/210 seems inappropriate. That would indeed be useful for telling apart e.g. an alpha-to-beta transition, or formation of coiled coils. However, for a helix-to-coil equilibrium, which is likely to dominate here, it will not be especially sensitive as demonstrated e.g. by BSA in the dry state.

We thank the reviewer for this comment. The use of ratios to measure structural transition is primarily to eliminate the effects of concentrations on the graph. It is clear from Figure 3A and Figure 3B that a structural transition occurs between the aqueous and the desiccated state. This is also very clear from the 222/210 ratio that we use (Figure 3C), for every construct other than BSA - which indeed does not seem to undergo a dramatic structural change in the desiccated state. We have clarified this now in the description of the results: “Using this metric, all LEAs and CAHS D display a clear increase in helical propensity upon being desiccated (Figure 3C). On the other hand, the helical propensity of BSA remains very similar to its hydrated state, indicating that no dramatic structural change took place (Figure 3C).

Minor comments:Figure 1F is not mentioned in the text.

We have included Figure 1F in the text.

Some technical details missing for SAXS experiments.

We thank the reviewer for pointing this out. We’ve added additional technical details to the main text, and directed readers to the methods for more information.

It is well known that BSA is in a monomer-dimer equilibrium and this is normally taken into account in data analysis as this is often a calibration sample.

We’ve calculated ΔΔGtrM→D for BSA, and correlated the resulting data with synergy. This can be found in Figure 5—figure supplement 1M and Figure 6—figure supplement 1I.

Line 247: "BSA, which comes from cows, which of course have no capacity for anhydrobiosis" - This seems like a rather strong statement without a reference. Did the authors consider reanimating beef jerky by soaking it in water?

This is a great idea, and we hope to assign this project to our next rotation student.

Minor suggestions for figures (that are generally very well done):Figure 1-4: Consider using the color scheme to indicate what the endogenous cosolutes are. Even though this info is in table one, it would still improve readability.

We have added the colored organismal icons for all figures in which the plain black ones were previously used, including supplementals.

Figure 4: consider adding some white space between the two concentration series of solutes to avoid being read as a single concentration series.

We have updated this figure to clearly separate each sample by osmolyte.

Figure 6H: Consider changing the colors for Betaine and CAHS D, so they are easier to distinguish. They are hard to tell apart on a printout.

We have adjusted the colors for betaine and CAHS D.